# Spatial variability of air pollutants in a megacity characterized by mobile measurements

Reza Bashiri Khuzestani[1,2,†], Keren Liao[1,†], Ying Liu[1], Ruqian Miao[1], Yan Zheng[1], Xi Cheng[1], Tianjiao Jia[1], Xin Li[1], Shiyi Chen[1], Guancong Huang[1], Qi Chen[1,*]

[1]State Key Joint Laboratory of Environmental Simulation and Pollution Control, BIC-ESAT and IJRC, College .of Environmental Sciences and Engineering, Peking University, Beijing, 100871, China
[2]Now at: Faculty of Civil, Water and Environmental Engineering, School of Engineering, Shahid Beheshti University, Tehran, Iran
[†]Equal Contribution.

*Correspondence to*: Qi Chen (qichenpku@pku.edu.cn)

**Abstract.** Characterization of the spatial distributions of air pollutants on an intracity scale is important for understanding local sources, secondary formation, and human exposure. In this study, we conducted in situ mobile measurements for the chemical composition of fine particles, volatile organic compounds (VOCs), oxygenated VOCs (OVOCs), and common gas pollutants in winter in the megacity of Beijing. The spatial patterns of these pollutants under different pollution conditions were investigated. During the non-haze days, all pollutants showed significant spatial variability. Large spatial variations of secondary species including OVOCs and secondary aerosol species highlight the chemical heterogeneity. In particular, the inorganic chemical composition of fine particles varied greatly on the 65-km urban highway, suggesting a wide range of particle neutralization in the megacity of Beijing. Localized sources such as vehicle, cooking, and industry emissions led to hot spots and nonuniform distributions of primary pollutants in the city. The spatial heterogeneity of air pollutants under less-polluted conditions calls a future need of using fine-resolution models to evaluate human exposure and to develop pollution control strategies. During the haze day, the spatial variabilities of secondary gaseous and particulate pollutants were largely reduced, explained by both of the elevated urban background from polluted air mass and the enhanced secondary formation by elevated precursor concentrations and heterogeneous or aqueous pathways. Although localized primary emissions were accumulated under stagnant haze conditions, the chemical composition of fine particles became relatively homogeneous because of the predominant secondary contributions. An uniform spatial pattern of particle neutralization was observed. The concentrations of hydrocarbons and less-oxygenated OVOCs showed good positive spatial-temporal correlations during the haze day, while the concentrations of more-oxygenated OVOCs showed good positive correlations among themselves but moderate negative correlations with the concentrations of hydrocarbons, less-oxygenated OVOCs, and aerosol species. Our results indicate a spatial chemical homogeneity on the megacity scale to promote efficient SOA production under haze conditions.

# 1 Introduction

Air quality degradation has become a serious problem in developing countries (Huang et al., 2014; Khuzestani et al., 2017). Cities in northern China suffer from severe haze events, especially in winter (Huang et al., 2014). High concentrations of volatile organic compounds (VOCs) and nitrogen oxides ($NO_x = NO + NO_2$) contribute to high atmospheric oxidation capacity to produce ozone ($O_3$) and secondary aerosol (Wang et al., 2017; Lu et al., 2019b). The mass concentrations of particles having aerodynamic diameter less than 2.5 μm ($PM_{2.5}$) may reach up to several hundreds of μg m$^{-3}$, and the $O_3$ concentrations may exceed 100 ppbv in megacities. Both of high exposures of $PM_{2.5}$ and $O_3$ are of great concern to human health (Liu et al., 2016; Bell et al., 2006).

Tremendous efforts have been made to characterize air pollutants in populated urban areas in China (Li et al., 2017; Wang et al., 2017; Quan and Jia, 2020). In most studies, stationary measurements were conducted to obtain the temporal variations of air pollutants for the analysis of their sources and formation mechanisms. Sulfate, nitrate, ammonium, and organic aerosol (OA) are the major chemical components of $PM_{2.5}$ in cities in China (Li et al., 2017). Their mass concentrations show decreases nationwide since 2013 as a result of the emission reduction of gaseous precursors such as sulfur dioxide ($SO_2$), $NO_x$, and VOCs under clean air actions as well as the changes of meteorological conditions (Zhang et al., 2019). Characterization of VOCs and oxygenated VOCs (OVOCs) is crucial for understanding the formation of $O_3$ and secondary OA (SOA). Alkanes, aromatics and OVOCs are abundant in urban areas in China (Mozaffar and Zhang, 2020; Li et al., 2016; Guo et al., 2017). These organic compounds may come from a variety of anthropogenic sources such as vehicles, industries, solvent usage, coal and biomass burning, and household use of volatile chemical products etc. In Beijing, the contribution of SOA to $PM_{2.5}$ has significantly increased since 2018, while the contribution of primary OA (POA) from coal and biomass (biofuel) combustion and cooking emissions have decreased (Lei et al., 2021).

On an intracity scale, local emissions, advection, and boundary layer evolution affect the spatial distributions of air pollutants. Tower measurements have shown greater vertical gradients of particle concentration and composition during clean periods than during haze periods in Beijing, which is attributed to the influence of both physical (e.g., regional transport, mixing layer height, and inversions etc.) and chemical processes (e.g., gas-particle partitioning and aqueous processing) (Zhou et al., 2018). The VOC concentrations also show significant vertical gradients in cities, affected by the transport of air masses, vertical diffusion, and species reactivity (Sun et al., 2018; Mo et al., 2020). Grid sampling and simultaneous measurements at multiple sites have shown that the surface concentrations of VOCs and $PM_{2.5}$ chemical components can vary significantly on sparse spatial scales because of different source contributions in metropolitan areas in China (Chen et al., 2020; Song et al., 2021). Such measurements are however limited. The spatial variability of air pollutants remains quite unclear, limiting our understanding of source distribution, haze development, and human exposure.

Mobile laboratory equipped with fast response in situ instruments have been widely used in Europe and United States to better characterize the spatial distributions of air pollutants (Kolb et al., 2004). For example, mobile measurements in Pittsburgh and

Zurich indicate different spatial variability of submicron particle composition (Tan et al., 2014; Gu et al., 2018; Ye et al., 2018; Mohr et al., 2011; Elser et al., 2016). While primary carbonaceous sources led to the large spatial heterogeneity of the particle composition in Pittsburgh, a rather uniform distribution was observed in the metropolitan area of Zurich especially during thermal inversions over the Swiss plateau. In China, mobile measurements have been conducted to obtain the spatial distributions of $SO_2$, $NO_x$, carbon monoxide (CO), and black carbon (BC) as well as the column densities of $NO_2$ and $SO_2$ for the evaluation of regional transport routes in North China Plain (NCP) and the sources of these pollutants (Wang et al., 2009; Wang et al., 2011; Zhu et al., 2016; Wu et al., 2018). Online mass spectrometers have also been used on the mobile measurements of VOCs and $PM_{2.5}$ chemical composition, although the spatial variabilities of these pollutants have not yet been systematically explored (X. Wang et al., 2021; Liang et al., 2020; Liao et al., 2021).

In this study, we conducted in situ sampling with gas analyzers and mass spectrometers on a newly developed mobile laboratory in winter in Beijing. Spatial distributions of the mass concentrations of non-refractory $PM_{2.5}$ (NR-$PM_{2.5}$) components (i.e., OA, nitrate, sulfate, ammonium, and chloride), VOC and OVOC species, and common gas pollutants such as $SO_2$, CO, NO, $NO_2$, and $O_3$ were obtained. The spatial variabilities of these gaseous and particulate pollutants under different pollution conditions were investigated to provide new insights into urban pollution.

## 2 Methods

On-road measurements were conducted by the PKU mobile laboratory on the 4th Ring Road of Beijing on 7-21 November 2018. The 4th Ring Road is a 65-km-long urban highway that passes through residential, commercial and services, park, and transportation areas in the megacity (Figure 1a). Industrial facilities are mostly located in the south of Beijing and outside of the 4th Ring Road. The average traffic volume on the 4th Ring Road is about $1.2 \times 10^4$ vehicles per hour (Figure 1b), and over 90% of the fleet consisted of light duty gasoline vehicles (LDGVs) that meet National Stage III to V emission standards (Wu et al., 2017; Deng et al., 2020; Lv et al., 2020). Our measurements started at around 9:00 a.m. and went one cycle after another till 4:00 p.m. to avoid rush hours. Each cycle took approximately 70 min with a mean speed of 60 km h$^{-1}$. Self-contamination from the vehicle exhaust should be negligible for a vehicle speed over 30 km h$^{-1}$ (Liao et al., 2021). The data that were collected at a speed of less than 30 km h$^{-1}$ during occasional traffic jams have been excluded from the analysis. Additionally, online measurements of gaseous and particulate pollutants were conducted at a roof station in the PKU campus (39.99 °N, 116.32 °E) as a reference. Temperature, RH, barometric pressure, wind speed, and wind direction were acquired by a Met One weather station (083E, 092, 010C, and 020C). Gas pollutants were measured by Thermo Scientific instruments, including CO (48i-TL), NO-$NO_2$-$NO_x$ (42i-TL), $SO_2$ (43i-TL), and $O_3$ (49i-TL). $PM_{2.5}$ mass concentrations were measured by a tapered element oscillating microbalance monitor (Thermo, TEOM 1400A). Non-refractory chemical components of submicron particles (NR-$PM_1$) were measured by an Aerodyne long time-of-flight aerosol mass spectrometer (LTOF-AMS). The roof site is located

between the 4th and the 5th North Ring Roads, representing a typical urban background environment in Beijing(Zheng et al., 2021).

A suite of instruments was deployed in the mobile laboratory. Detailed information about the sampling system, instrument operation, calibration, and data analysis are provided in Sect. A of the Supplement. Gas and particle inlets were installed at the top front of the vehicle (i.e., about 3.4 m above ground) to sample on-road air that was influenced by diluted vehicle exhaust (Figure S1 in the Supplement). Gas pollutants were detected by gas analyzers including $NO_2$ (Teledyne, T500U), NO-$NO_x$ (Ecotech, EC9841A), $SO_2$ (Ecotech, EC9850A), CO (Ecotech, EC9830A), and $O_3$ (Ecotech, EC9810A) with a time resolution of 2 s. The chemical composition of NR-$PM_{2.5}$ was measured by an Aerodyne time-of-flight aerosol chemical speciation monitor (TOF-ACSM) equipped with $PM_{2.5}$ lens and capture vaporizer. The time resolution was 40 s, corresponding to a spatial resolution of ~0.7 km for a driving speed of 60 km $h^{-1}$. The mass resolution was about 400. A collection efficiency of about unity was used, and the uncertainty of mass quantification was ~30% (Zheng et al., 2020). Positive matrix factorization (PMF) analysis was performed on the unit-mass-resolution OA mass spectra for mass-to-charge ratio ($m/z$) of 12 to 200 by using the Igor PMF evaluation tool (PET, version 3.00B). Details are given in Table S1, Figures S2-S5, and Section A3 of the Supplement. A total of five OA factors has been resolved including hydrocarbon-like OA (HOA), cooking-related OA (COA), and three oxygenated OA (OOAs). The common OA factors related to biomass burning (BBOA) or coal burning (CCOA) were not resolved in this data set. They were likely mixed with HOA because of the similarity in the CV spectra (Zheng et al., 2020). Their contributions to OA were expected to be small because of the stringent emission control in NCP in recent years (Zheng et al., 2020; Duan et al., 2020). Consistently, the PTR-MS data showed low concentrations of acetonitrile (i.e., 0.15 $\pm$ 0.20 ppbv for non-haze periods and 0.60 $\pm$ 0.22 ppbv for the haze periods) during the mobile campaign. By comparison, the average concentrations of acetonitrile are usually greater than 0.7-1 ppbv in the winter of Beijing when biomass and coal burning contributes significantly (Huangfu et al., 2021; Shi et al., 2020).

VOCs and OVOCs were detected by an Ionicon proton transfer reaction-quadrupole ion guide time-of-flight mass spectrometer (PTR-QiTOF) with a time resolution of 2 s. The overall quantification uncertainties were less than 20% for calibrated species, and 19% or 33% for uncalibrated species with known or unknown reaction rate constants ($K_{ptr}$), respectively (Huang et al., 2019). Table S2 in the Supplement lists the $K_{ptr}$ values, the mean concentrations, and the tentative categorization of VOCs and OVOCs measured herein (Yuan et al., 2017). A moving average window of 10 (i.e., 20 s) was applied to the entire data set for the VOC and OVOC species. Baseline concentrations for each 2-s point in the 20-s smoothed data were calculated as the 5th percentile concentration within a rolling window of 60 (i.e., 120 s) to represent the well-mixed urban background. Figure S6 in the Supplement shows the time series and the baselines of selected VOC and OVOC concentrations during two typical runs on the 4th Ring Road. We calculated the peak fraction (PF) parameter as 1 minus the ratio of the mean baseline value to the observed mean concentrations to indicate the contributions to the mean observed concentrations above the mean local baseline (Apte et al., 2017). In addition, we computed a batch of 12-hour backward trajectories for the height of 3 m around the 4th Ring Road from the Hybrid Single-Particle Lagrangian Integrated Trajectory (HYSPLIT) model to investigate the influence

of regional transport on our measurements during the haze day (Stein et al., 2015). The start time of the trajectory analysis was set at 9:00 a.m., and the trajectory calculations were repeated every 1 hour until 4:00 p.m. (Figure S7 in the Supplement).

## 3 Results and Discussion

### 3.1 Spatial distribution and variability during the non-haze days

Figure S8 in the Supplement shows the time series of meteorological parameters and mass concentrations (or mixing ratios) of air pollutants measured at the PKU roof station during the mobile campaign. We classified haze ($\geq 75$ μg m$^{-3}$) and non-haze ($< 75$ μg m$^{-3}$) days based on the daily mean PM$_{2.5}$ mass concentrations measured at the roof site. The mobile measurements covered from 9 a.m. to 4 p.m. for 8 non-haze days and 1 haze day. Apte et al. (2017) suggest that a small number of drive days (i.e., $< 5$ days) may result in a biased data set to represent the long-term spatial patterns. Although we have limited data (1 day)

to represent the weekend or haze conditions, some of the key characteristics of air pollutants from the mobile measurements are consistent with previous understanding about weekend effects and haze evolution in Beijing (Table S3 in the Supplement). For instance, lower mass concentrations of aerosol species presented at weekend than on weekdays in winter, similar to previous findings (Sun et al., 2015). NO$_x$ and VOC concentrations were lower at weekend than on weekdays because of heavier traffic loads and anthropogenic activities. By contrast, O$_3$ concentrations were greater at weekend, indicating enhanced

photochemical production (Wang et al., 2014; Liu et al., 2020). The severe haze event occurred on 13-14 November 2018 under stagnant weather conditions with occasional shifts between northerly and southerly winds (Figure S8). The PM$_{2.5}$ concentrations increased rapidly to nearly 300 μg m$^{-3}$ during that event. Relative humidity (RH) increased and remained above 70% on 14 November 2018, which was often observed in the later stage of severe winter haze in NCP in China (Sun et al., 2016). The elevated RH can lead to a large increase of aerosol liquid water content which may promote aqueous or

heterogeneous chemistry to enhance the formation of secondary species (Peng et al., 2021). Indeed, the mass concentrations of sulfate, nitrate, and OA increased greatly, and nitrate became the most abundant component during the haze day. Moreover, the meteorological conditions and pollutant concentrations were similar to previous findings in Beijing (Zheng et al., 2021). The aerosol compositions measured by the mobile laboratory within a distance of 1.5 km from the site were also similar to the roof-site observations (Figure 1c-d), taking into account the potential difference between PM$_1$ and PM$_{2.5}$ (Sun et al., 2020).

Figure 2 shows the spatial distributions of the mean mixing ratios of gaseous pollutants. During the non-haze days, gas pollutants all showed significant spatial variability. The mean mixing ratios varied by 2-3 times for SO$_2$, CO, NO, NO$_2$, and O$_3$ and by 4 to 21 times for VOC subgroups on the 4th Ring Road. For SO$_2$, the mean mixing ratios ranged from 3.7 to 7.1 ppbv. The lack of hot spots in the spatial pattern indicates a minor influence of on-road or near-road (e.g., off-road engines) sources. The overall low concentrations of SO$_2$ are consistent with the predominant contribution of regional transport as a

result of the phasing out of coal use in Beijing including power plants, coal boilers, and residential stoves (Zheng et al., 2018; Ge et al., 2018). High SO$_2$ concentrations presented in the northwest segment, while the low concentrations were in the north

segment. In summer, the southeast prevailing wind can lead to higher concentrations of $SO_2$ in the southeast areas (Wang et al., 2011; Zhu et al., 2016). The northwest prevailing wind in winter brings however usually clean air masses that are far away from the NCP regional sources (Figure 1c). The relatively high $SO_2$ mixing ratios in the northwest segment was perhaps associated with residential coal burning near the mountain area where the replacement of coal has been lagged compared with the plain area.

For CO, the mean mixing ratio of 1.5 $\pm$ 0.7 ppmv was about three times greater than the urban background level (0.5 $\pm$ 0.3 ppm at the PKU roof site), indicating significant contributions of localized sources during the non-haze days. Vehicle emissions are the main local source of $NO_x$ and CO in Beijing (Qi et al., 2017). The spatial distribution of $NO_x$ was generally consistent with that of CO, showing relatively high mixing ratios in the west and east segments of the 4th Ring Road (Figure S9 in the Supplement). Differences remained in the southwest corner where CO showed high concentrations but $NO_x$ did not, which was perhaps related to non-vehicle sources or different fleet composition. On-road $NO_2$ can be contributed by direct tailpipe $NO_2$ emissions, NO titration, and urban background. The mean mixing ratios of on-road $O_x$ (114.1 ppbv) were much greater than the roof-site mean ratio (48.7 ppbv) during the non-haze periods, indicating a possible contribution of direct tailpipe $NO_2$ emissions, although tailpipe $NO_2$ emissions for LDGVs of National Stage III to V are expected to be low (Wu et al., 2017). On-road NO titration can also be strong (Yang et al., 2018), considering that the spatial patterns for $O_3$ and NO were anti-correlated (Pearson $r$ = -0.43) and the mean mixing ratio of on-road $O_3$ (11.2 $\pm$ 2.2 ppbv) were over 2 times lower than the roof-site observations (25.7 $\pm$ 12.8 ppbv). Quantitative analysis on their relative contributions to on-road $NO_2$ is however difficult because the roof site was located in the upwind direction. Overall, the spatial pattern of $NO_x$ was consistent with the bottom-up emission inventory for (1) the nonuniform vehicle emissions on the 4th Ring Road and (2) high concentrations in the east segment of the 4th Ring Road where the traffic volume was high (Figure 1b and Figure S10 in the Supplement).

The spatial distributions of VOCs varied by species. LDGVs emit aromatic hydrocarbons (e.g., benzene, toluene, ethylbenzene, and xylenes) and some oxygenated species (e.g., acetaldehyde) (Gentner et al., 2017). Solvent use and residential solid-fuel combustion are the other localized sources for hydrocarbons (L. W. Wang et al., 2021). The on-road concentrations of such primary VOCs were much greater than the concentrations measured at the urban and suburban sites in Beijing (Table S2). During the non-haze days, the tentatively-assigned group of hydrocarbons showed the greatest spatial variability (by 21$\times$) among VOCs, whereas the groups of aldehydes/ketones and acids/anhydrides showed less variability (by 4$\times$ and 9$\times$, respectively) (Table S2). Secondary production is expected to contribute greatly to the latter two groups (L. W. Wang et al., 2021). The toluene-to-benzene concentration ratio (T/B) has been widely used to differentiate the VOC sources. Some hot spots in the spatial pattern of hydrocarbons (e.g., high concentrations in the northwest corner and the northeast segment) were plausibly contributed by transient plumes of vehicle exhaust (Figure S10). The T/B ratios was ~1 in the northwest corner and ~2 in the northeast segment (Figure S9), suggesting perhaps different fleet composition (Mo et al., 2016). Industry and solvent emissions have greater T/B ratio than vehicle emissions, whereas emissions from biomass or solid fuel burning or aged air mass typically have lower T/B ratios (Song et al., 2021; and references therein). Some other high T/B ratios (~3) in the south

and southeast segments might be affected by industry emissions in the south of Beijing (Figure 1a). The spatial patterns of hydrocarbons (or benzene and toluene) were different from that of NO or $NO_x$, explained by the potential differences in fleet composition and urban background contributions as well as the chemical conversion rate.

Figure 3 shows the spatial distributions of the mean mass concentrations of key $PM_{2.5}$ components. During the non-haze days, the mean concentrations of OA, nitrate, sulfate, and ammonium on the 4th Ring Road varied by 9 to 13 times. The coefficients of variation (CV, the standard deviation divided by the mean) of the four components and chloride ranged from 38% to 84%, indicating strong variability on the 24×17 km intracity scale (Table S4 in the Supplement). The spatial variability of aerosol species was similar to that of VOCs but greater than that of other gas pollutants. OA was the major contributor to the NR-$PM_{2.5}$ mass (Table S3). Its concentration varied by 9 times on the 4th Ring Road. The spatial variability of the OA mass was attributed to both of POA and SOA. As shown in Figure 4, the mass concentrations of POA factors such as HOA and COA varied from 0 to 20 μg m$^{-3}$ with hot spots in different segments of the 4th Ring Road. The 40-s $PM_{2.5}$ measurements by TOF-ACSM may roughly represent a maximum area of 0.16 km$^2$ (for a mean wind speed of 6 m s$^{-1}$ and wind direction perpendicular to the mobile path) upwind when the mobile laboratory was run on the 4th Ring Road by cycles. The HOA hot spots are generally consistent with the locations where the driving speed was relatively low (i.e., perhaps high traffic volume). We use the driving speed to indicate the traffic volume because the real-time traffic volume data weren't available (Figure S10 in the Supplement). The COA hot spots are consistent with the places where the 4th Ring Road passes through sparsely located residential areas (Figure 1a).

Moreover, the mass concentrations of the sum of OOAs varied from 0 to 15 μg m$^{-3}$. Local photochemical production of SOA is a significant source of OA in Beijing in winter, although the solar radiation is reduced (Duan et al., 2020; Lu et al., 2019a). The photochemical production depends on the distributions of SOA precursors and oxidants. In the northwest corner where hydrocarbons showed high concentrations, the OOA mass loadings were indeed high. Because the majority of the SOA precursors (i.e., intermediate volatility and semivolatile organic species from anthropogenic sources) were not measured by the PTR-Qi-ToF (Liao et al., 2021; Miao et al., 2021), it is difficult to investigate more about the OOA sources. The measurements in Pittsburgh also showed a significant spatial heterogeneity of primary carbonaceous components such as HOA, COA, and BC (Gu et al., 2018). Less spatial variability presented for OOAs in the Pittsburgh study. The OA mass loadings in Pittsburgh were however much less than the loadings in Beijing. The SOA formation can be significantly more efficient and complicated under conditions of high oxidative capacity and abundant precursors in Beijing than in Pittsburgh (Lu et al., 2019a; Li et al., 2021; Yang et al., 2019). Non-perfect separation of POA from SOA by the PMF analysis may also lead to misplaced spatial variability in OOA. For example, the CV-based PMF analysis may overestimate the SOA mass comparing with the traditional AMS analysis (Zheng et al., 2020), which may lead to inappropriate attribution of POA to SOA and thus more spatial variability in SOA. Uncertainty remains in the mass separation of POA and SOA (Sect. A3 of the Supplement).

Among the inorganic aerosol species, chloride has the highest CV (Table S4). The mean concentration of chloride varied from below detection limit (0.04 μg m$^{-3}$) to 0.6 μg m$^{-3}$. Although the TOF-ACSM equipped with capture vaporizer may

underestimate the chloride mass concentrations by a factor of 2, its contribution to the mass and spatial variability of PM$_{2.5}$ were negligible. The main source of chloride in winter in Beijing is coal burning by regional transport, which should not lead to high spatial variability. The cause of such a high spatial variability of chloride during the non-haze days remains unknown. One hypothesis is the resuspended particles from the road (Chen et al., 2012). The spatial variability of sulfate, nitrate, and ammonium were similar in terms of CV and the concentration variations. The spatial pattern of ammonium however showed a large heterogeneity. The molar ratio of $n_{\mathrm{NH_4^+}} : (2 \times n_{\mathrm{SO_4^{2-}}} + n_{\mathrm{NO_3^-}})$ ranged from ~ 0.5 to 1.7 (Figure 5), suggesting a wide range of particle neutralization in winter in Beijing during the non-haze days. In Beijing, the aerosol mass can be sensitive to either ammonia or the availability of nitric acid (Nenes et al., 2020). Traffic and agriculture (e.g., livestock) emissions are the main sources of ammonia in Beijing in winter (Sun et al., 2017; Pan et al., 2018). The former depends on vehicle type and volume and hence may vary greater on the Ring Road (Sun et al., 2017). The concentrations of nitric acid depend on the concentrations of NO$_x$ and hydroxyl radical (OH), which may also vary spatially. Besides, the distribution of solar radiation on the surface is nonuniform because of the shades of buildings and occasional cloud cover. Therefore, the spatial heterogeneity of inorganic composition can be caused by different thermodynamic regimes in the megacity. The inorganic composition have less spatial variability in Pittsburgh than in Beijing (Gu et al., 2018). A possible explanation is that the formation of ammonium nitrate is sensitive to ammonia only in Pittsburgh (Nenes et al., 2020). The heterogeneity of inorganic composition can be seen in each of the single cycle on the 4th Ring Road (Figure S11 in the Supplement).

**3.2 Spatial distribution and variability during the haze day**

The spatial patterns of air pollutants were very different under different pollution conditions. As shown in Figure 2, the mean mixing ratios of SO$_2$ during the haze day were lower than the non-haze day case (Table S3), explained by the large conversion of gaseous SO$_2$ to sulfate by both of the photochemical and aqueous processes under severe haze conditions (W. G. Wang et al., 2021). The mean mixing ratios of CO were however greater than the non-haze day ratios, indicating accumulated pollution. CO and SO$_2$ show relatively high concentrations in the east and north segments of the 4th Ring Road during the haze day. The haze in NCP is usually developed regionally, meaning that the polluted air mass travels and would become more polluted when it suspends in urban areas to accumulate local emissions and secondary production under stagnant conditions. The 12-hour backward trajectories around the 4th Ring Road show that the air masses were mostly from the east in the morning, started shifting towards south at ~12:00 p.m., and then were mainly originated from the south around 4:00 p.m. (Figure S7 in the Supplement). Southwesterly, southeasterly, and easterly fluxes have been recognized as regional pollution sources to Beijing (Chang et al., 2019). The spatial patterns of SO$_2$ and CO are thus consistent with the transport routes, and the low concentrations in the southwest suggest a possible edge of the polluted air mass (Peng et al., 2021). For NO, the mean mixing ratio at the PKU roof site during the haze day (46.0 ±19.8 ppbv) was much greater than that observed for the non-haze days (14.7 ±17.4 ppbv), but was still much lower than the on-road mixing ratios because of the titration. The spatial patterns of NO were similar for the non-haze and haze days, except the slightly greater concentrations in the north segment during the haze day. The mean NO$_2$ concentrations (69.7 ±26.5 ppbv) were lower during the haze day than during the non-haze days (102.9 ±37.3 ppbv).

The corresponding PKU roof-site mean concentrations of $NO_2$ was $63.0 \pm 8.8$ ppbv, which is similar to the on-road concentrations. This indicates a potentially high urban background contribution to on-road $NO_2$ concentrations during the haze day, which is plausibly greater than its contribution during the non-haze days. The spatial pattern of $NO_2$ was determined by both on-road titration and urban background (similarly for $O_3$) during the haze day, and thus the spatial patterns of $NO_2$ and $O_3$ were different from those for the non-haze days.

Overall, the spatial variabilities for $SO_2$, CO, NO, $NO_2$, and $O_3$ were similar during the non-haze and haze days (i.e., varied by $2$-$4\times$ with similar CV values) (Table S4). The spatial variability for VOCs was greatly reduced during the haze day (by $2$-$3\times$ in contrast to $4$-$21\times$ during the non-haze days), especially for the OVOCs. The spatial patterns of benzene and toluene during the haze day were different from those during the non-haze days with lower T/B ratios. The low ratios suggest aged air mass during the haze day, which is consistent with the influence of regional pollution (de Gouw et al., 2005). The spatial patterns of OVOCs are similar to each other except for the OVOCs having 3-4 oxygen atoms in their formulae (see more discussion in Section 3.3). The calculated PF for VOCs ranged from 11-67% (median) (Figure 6). High PF values were found for hydrocarbons and some OVOCs (e.g., $C_8H_8$, $C_{10}H_8$ and $C_4H_4O$), indicating a major contribution of transient localized sources (e.g., traffic, industrial facilities) to these species. The time series of these so-called primary species showed low baselines and sharp peaks (Figure S6). By contrast, OVOCs (i.e., with 2 or more oxygen in their formulae) that were typically considered as secondary species had low PF values (median: 11-16%) and elevated baseline contribution from photochemistry. Significance tests indicate greater PF values for the primary species during the non-haze days, meaning that the localized sources contributed more to the measured concentrations during the non-haze days than during the haze day ($p < 0.001$). During the haze day, the localized emissions should be accumulated near the source under stagnant conditions. Indeed, the peak concentrations of primary VOC species were significantly greater (e.g., $\sim$2-$4\times$ for $C_6H_6$ and $C_7H_8$) (Figure S6). The lower PF values (by 30% for $C_6H_6$ and $C_7H_8$) during the haze day were caused by much more elevated baselines (e.g., $\sim$9$\times$ for $C_6H_6$ and $C_7H_8$) that represent urban background affected by polluted air mass from regional transport plus gradually mixed local emissions. The mean VOC concentrations at the PKU roof site increased for about 2 times during the haze day, which agrees with the elevated baselines (Table S3 and Figure S12 in the Supplement).

The haze-day mean concentrations of aerosol species increased by about 5 times on the 4th Ring Road (Figure 3). Similar to OVOCs, the spatial variability of aerosol species were greatly reduced with CV values of $< 35\%$ vs. 38-84% for the non-haze days. Hot spots of HOA and COA became more evident, which is consistent with the less-dispersed primary emissions under stagnant conditions (Figure 4). The OOA mass increased most and showed a spatial pattern that were consistent with inorganic species. Unlike the non-haze days, the spatial pattern of ammonium was similar to those of sulfate and nitrate. The chemical composition of $PM_{2.5}$ became relatively homogeneous during the haze day. The molar ratio of $n_{NH_4^+} : (2 \times n_{SO_4^{2-}} + n_{NO_3^-})$ was $\sim$1 (Figure 5), suggesting an uniform neutralization pattern over the city during the haze day. Although stagnant conditions facilitate the accumulation of localized emissions, polluted air mass leads to elevated urban background for both primary and secondary pollutants. Previous studies suggest that over 60% of the $PM_{2.5}$ mass in Beijing can be contributed by regional

transport during the winter haze event (Sun et al., 2014; Wu et al., 2021). Secondary formation can also be enhanced because of the elevated precursor concentrations during the haze day and heterogeneous and aqueous pathways that occur at the high-RH haze stage. Our results indicate a spatial chemical homogeneity perhaps in a regional scale in terms of particle formation (Sun et al., 2016; Chen et al., 2020). The uniform spatial distributions of PM composition under haze conditions are similar to the observations in the metropolitan area of Zurich when thermal inversions occur over the Swiss plateau and secondary pollution was built up regionally (Mohr et al., 2011), highlighting stagnant metrological conditions as one of the key drivers of the chemical homogeneity.

## 3.3 Chemical homogeneity of OVOC formation under haze conditions

Figure 7 shows the spatial-temporal variations of the concentrations of the detected VOCs and OVOCs measured during the non-haze and haze days. The concentrations of ∑hydrocarbons (i.e., aromatics and alkenes) were the highest in the morning (9:00 a.m. to 11:00 a.m.) for both non-haze and haze periods, which may be explained by shallow boundary layer, rush-hour traffic emissions, and slow chemical removal (de Gouw et al., 2009). The on-road measurements of hydrocarbons were largely affected by instantaneous vehicle plumes. The traffic volume in Beijing is usually less around noon than in the afternoon (Wu et al., 2017). Consistently, the concentrations of hydrocarbons were greater with larger variations in the afternoon (2:00 p.m. to 4:00 p.m.) than in the earlier period (11:00 a.m. to 2:00 p.m.), indicating more vehicle plumes were captured by the mobile measurements in the afternoon. Under non-haze conditions, the spatial variability of hydrocarbons varied significantly during the day. Their CV values were high in the morning and low in the afternoon. It is likely that photochemistry and better mixing conditions in the afternoon smoothed out some of the spatial variability caused by the on-road vehicle emissions (Mellouki et al., 2015; Karl et al., 2018). By contrast, their concentrations continued to decrease during the day under haze conditions, and the greater concentrations of ∑hydrocarbons than during the non-haze days were plausibly driven by the elevated urban background levels from regional pollution and less-dispersed vehicle plumes under stagnant conditions. Under haze conditions, the spatial variability of hydrocarbons was slightly greater in the afternoon, which was probably because of the change of regional transport direction. As shown in Figure S7, the backward trajectories suggest that the air masses gradually shifted from eastern to southern during the haze day. For the two categories of OVOCs, the non-haze temporal variations were similar to that of hydrocarbons. Their CV values indicate much smaller spatial variability of OVOCs in the morning rush hour (9:00 a.m. to 11:00 a.m.) than that of hydrocarbons, which is consistent with previous finding of the predominant contribution of secondary production to most OVOCs in Beijing (L. W. Wang et al., 2021). As the photochemical production proceeds during the day, the spatial variability of OVOCs increased and then decreased, similar to that of hydrocarbons. On the other hand, the haze-day temporal and spatial variations of OVOCs were similar to that of hydrocarbons (CV < 0.36), corresponding to the predominant contribution of regional transport to the on-road concentrations of these pollutants.

Figure 8 shows the correlation heatmaps of the concentrations of VOCs and OVOCs measured during the clean day and haze day from 9 a.m. to 4 p.m. on the 4th Ring Road. Only two days were considered in this analysis to avoid inconsistent source

profiles. Primary hydrocarbons showed good correlations with each other during the clean day (Figure 8a), which was consistent with predominant contribution of traffic emissions to their on-road concentrations. Secondary species like OVOCs and $PM_{2.5}$ were more regional and did not correlate with primary hydrocarbons (Figure S13 in the Supplement). They did not correlate with each other either, indicating a spatial chemical heterogeneity in OVOC and $PM_{2.5}$ formation. By contrast, the

haze-day heatmaps show significantly different patterns (Figure 8b). Primary hydrocarbons correlated well with each other (Pearson $r > 0.7$, $\alpha = 0.01$). They also showed good positive correlations with many less-oxygenated OVOCs (e.g., aldehydes and ketones with 1-2 oxygen atoms in their formulae (Table S2). These OVOCs may be contributed by both primary and secondary sources, and they can be formed and accumulated along the transport routes. The NR-$PM_{2.5}$ components also correlated with these VOCs and OVOCs (Figure S13). Such good spatial-temporal correlations are consistent with the

predominant contribution of regional pollution to high concentrations of hydrocarbons, less-oxygenated OVOCs, and $PM_{2.5}$ in Beijing during the haze event. Interestingly, more oxygenated OVOCs (e.g., anhydrides and acids with 2-4 oxygen atoms in their formulae) that were mainly contributed by secondary sources showed good positive correlations with each other ($r > 0.7$, $\alpha = 0.01$) but moderately negative correlations with hydrocarbons and less-oxygenated OVOCs as well as the NR-$PM_{2.5}$ components. These species include $CH_2O_2$ (tentatively assigned as formic acid), $C_2H_4O_3$, $C_3H_4O_3$, $C_4H_4O_4$, $C_4H_4O_3$, $C_5H_4O_3$,

and $C_5H_4O_2$. The spatial-temporal correlations for these more oxygenated OVOCs suggest similar more-oxygenated OVOC composition on the megacity scale during the haze day, which may indicate a homogeneous photochemical production as well as gas-to-particle partitioning. Figure 14 in the Supplement shows similar correlations that were observed in another haze event in January 2021.

High oxidation capacity in winter in NCP has been reported (Lu et al., 2019a; Slater et al., 2020). Secondary OVOCs can be

formed and accumulated along the transport routes. With a rather uniform distribution of hydroxyl radical which likely happens under winter haze conditions because of the thick cloud cover and increased particle scattering, the formation of more-oxygenated OVOCs from less-oxygenated OVOCs can be relatively homogeneous in the city. Although these more-oxygenated OVOCs may partition to the particle phase, the gas-to-particle partitioning were not expected to affect much the spatial variability of the more-oxygenated OVOCs because of the similar chemical composition of $PM_{2.5}$ and the narrow

loading range on the intracity scale under haze conditions. The secondary production can thus explain the negative correlations of the more-oxygenated OVOCs with their precursors (e.g., hydrocarbon and less-oxygenated OVOCs) and $PM_{2.5}$. Moreover, the gas-particle repartitioning caused by temperature difference may affect the distribution of more-oxygenated OVOCs. This effect is however expected to be minor considering the range of OA loadings, the vapor pressures of these more-oxygenated OVOCs, and the possible temperature elevation of 1-3 ℃ in the city (Chen et al., 2020; Wang et al., 2020; Yang et al., 2013).

Peroxyacetyl nitrate (PAN) can be detected as $C_2H_4O_3H^+$ in PTR-QiToF (i.e., $C_2H_4O_3$ in Figure 8). Peracetic acid and glycolic acid may also contribute to this ion, although their contributions are plausibly minor in urban environments (Hansel and Wisthaler, 2000; Kaser et al., 2013; Yuan et al., 2017). PAN is more sensitive to temperature and less stable than the other more-oxygenated OVOCs (Seinfeld and Pandis, 2016), which may explain its lower correlation coefficient in Figure 8b.

**4 Conclusions**

In this study, we conducted on-road mobile measurements of air pollutants including NR-PM$_{2.5}$ and its components, gaseous pollutants, VOCs, and OVOCs under different pollution levels in winter in Beijing. During the non-haze days, large spatial variabilities of these pollutants were observed, which can be largely attributed to the influence of localized sources, chemical reaction, and meteorological conditions. In particular, the inorganic composition of NR-PM$_{2.5}$ varied greatly, indicating a wide range of particle neutralization on the intracity scale. The spatial variabilities of HOA and COA were driven by vehicle and

cooking emissions as well as atmospheric dilution. The spatial patterns of VOCs and OVOCs varied by species. Hydrocarbons (i.e. mostly aromatic species) were greatly affected by vehicle emissions and showed greater PF values than OVOCs did. Secondary OVOCs showed the greatest spatial variability, indicating a large heterogeneity in their formation during the non-haze days. The spatial heterogeneity of the particle composition and VOCs may challenge the development of future pollution control strategies and accurate evaluation of human exposures in the megacity. On the other hand, the spatial variability of

secondary gaseous and aerosol pollutants contributions were largely reduced during the haze day. More-oxygenated OVOCs showed good positive correlation among themselves but moderate negative correlations with hydrocarbons, less-oxygenated OVOCs, and aerosol species, suggesting a spatially homogeneous chemical production as well as gas-to-particle partitioning on the megacity scale. Such a chemical homogeneity of OVOC formation may lead to efficient local production of semivolatile oxidation products that enhances the SOA production and the haze development.


*Data availability*. Data presented in this manuscript are available upon request to the corresponding author.

*Author contributions*. KL conducted the measurements and data analysis with the help of QC, RK, YL, YZ, XC, TJ, XL, SC, and GH. QC, RK, and KL wrote the manuscript.


*Competing interests*. The authors declare no competing financial interests.

*Acknowledgements*. This work was supported by the National Natural Science Foundation of China (91544107, 41875165, and 41961134034) and the 111 Project of Urban Air Pollution and Health Effects (B20009). The authors thank Theodore K.

Koenig for helpful discussions.

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

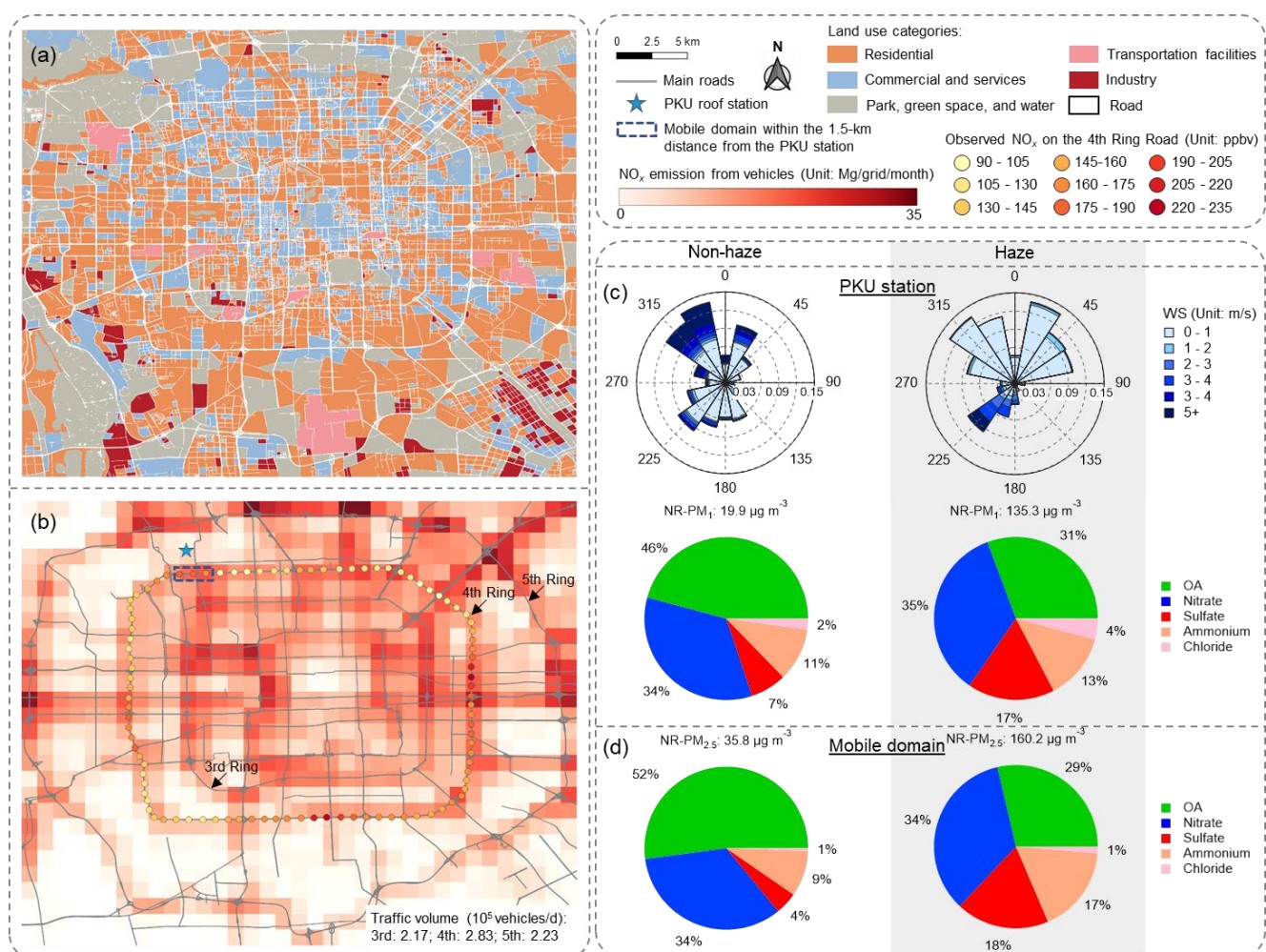

**Figure 1.** (a) Land use categories in the area of Beijing that were taken from the Mapping Essential Urban Land Use Categories in China (EULUC-China) database (Gong et al., 2020). (b) Monthly $NO_x$ emissions from vehicle provided by Yang et al. (2019) in November 2018 and the mean $NO_x$ concentrations observed by the mobile laboratory between 9 a.m. to 4 p.m. during the non-haze measurement period on the 4th Ring Road in Beijing. (c) Wind rose diagrams and the average NR-PM$_1$ composition measured by LTOF-AMS at the PKU roof site. (d) The average NR-PM$_{2.5}$ composition measured by the TOF-ACSM on the mobile lab on the 4th Ring Road within the distance of 1.5 km from the roof site. Base maps are provided by OpenStreetMap.

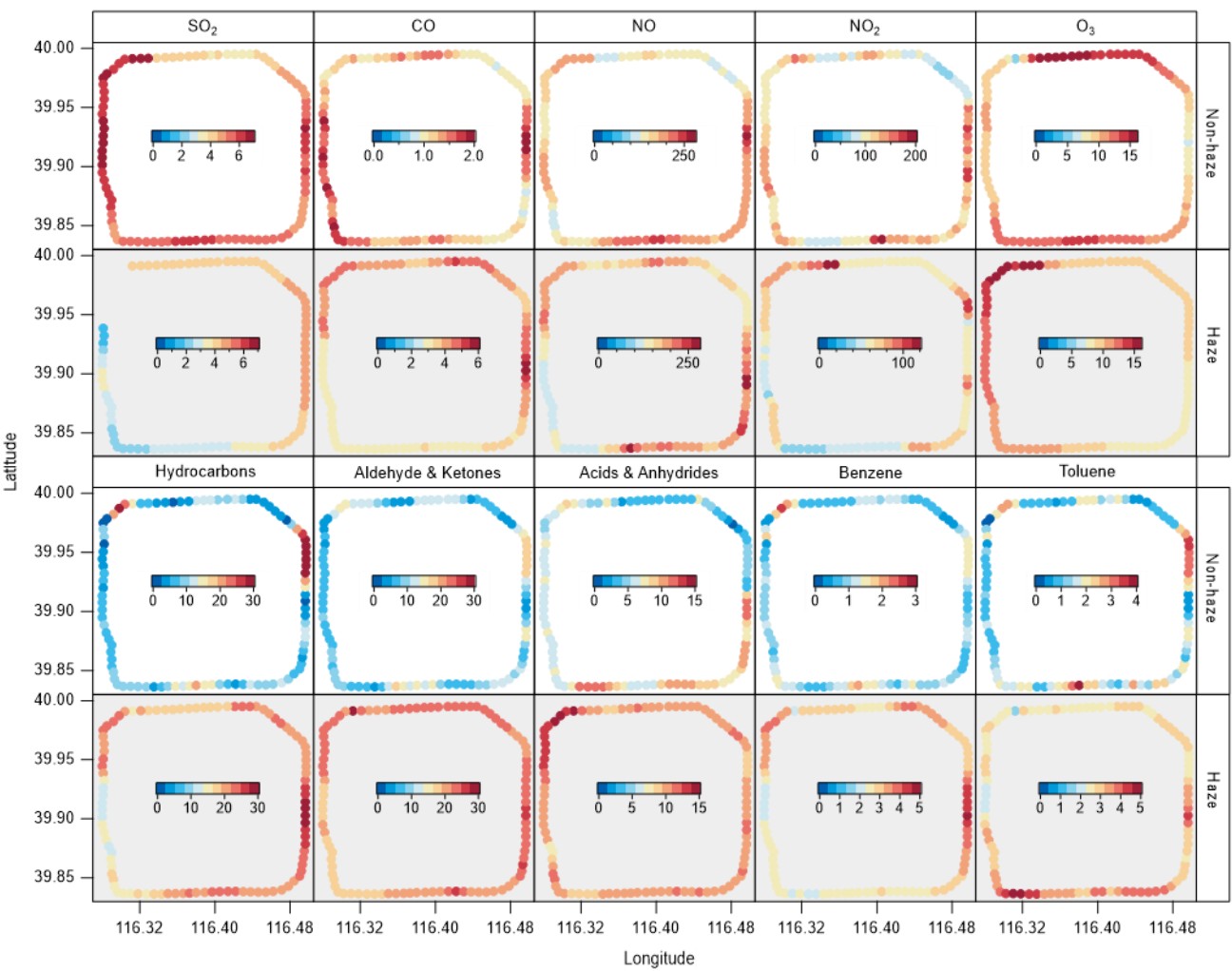

**Figure 2.** Spatial patterns of the mean mass mixing ratios (Unit: ppmv for CO and ppbv for others) of gas pollutants measured on the 4th Ring Road in Beijing. Data covered from 9 a.m. to 4 p.m. during the measurement period.

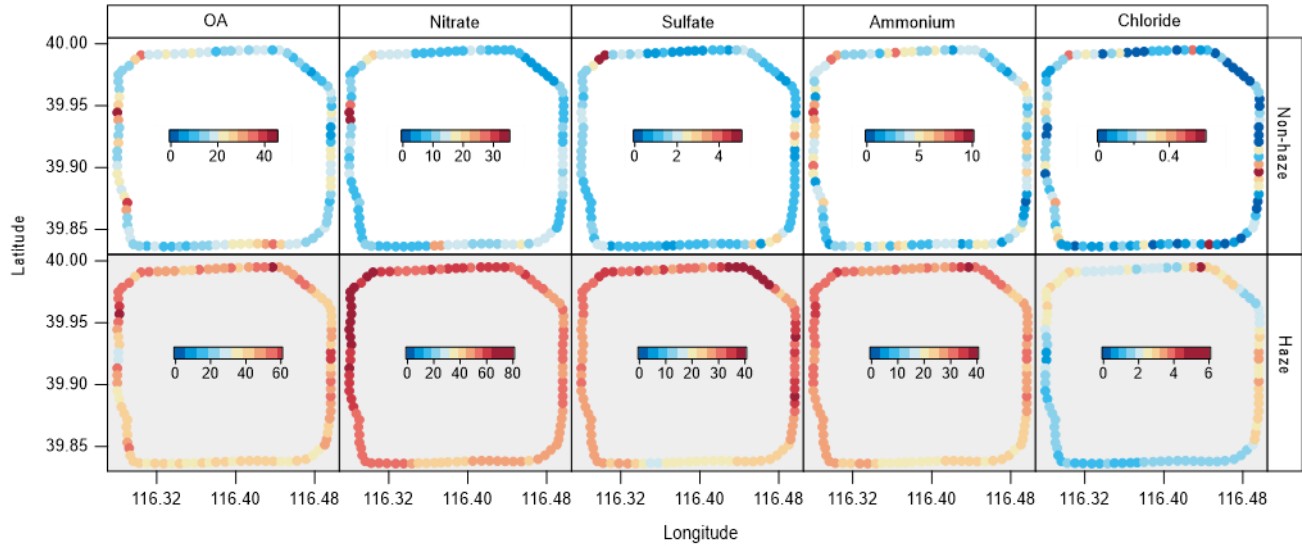

**Figure 3.** Spatial patterns of the mean mass concentrations (Unit: µg m$^{-3}$) of the key PM$_{2.5}$ chemical components measured on the 4th Ring Road in Beijing. Data covered from 9 a.m. to 4 p.m. during the measurement period.

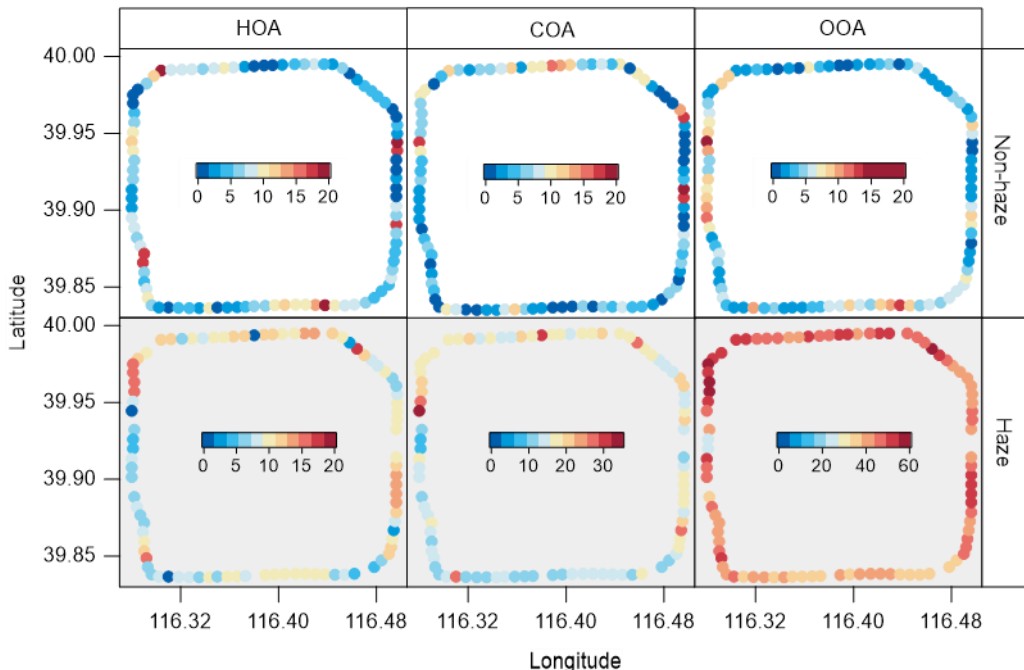

**Figure 4.** Spatial patterns of the mean mass concentrations (Unit: µg m$^{-3}$) of OA factors resolved by the PMF analysis of organic mass spectra obtained by the TOF-ACSM on the 4th Ring Road in Beijing. Data covered from 9 a.m. to 4 p.m. during the measurement period.

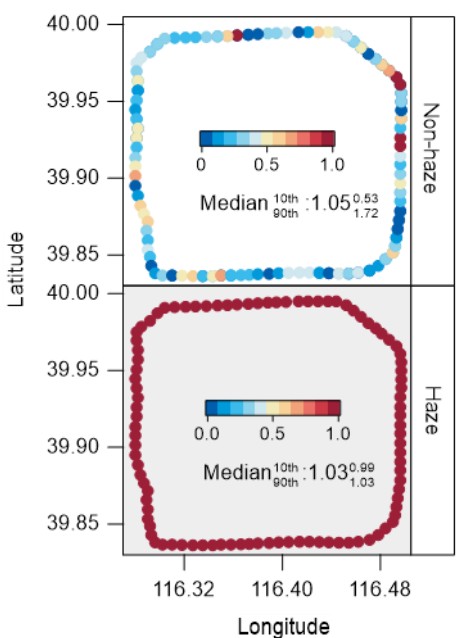

**Figure 5.** Spatial patterns of the molar ratio of $n_{\mathrm{NH_4^+}}:(2\times n_{\mathrm{SO_4^{2-}}}+n_{\mathrm{NO_3^-}})$ derived from the aerosol measurements on the 4th

Ring Road in Beijing. Data covered from 9 a.m. to 4 p.m. during the measurement period.

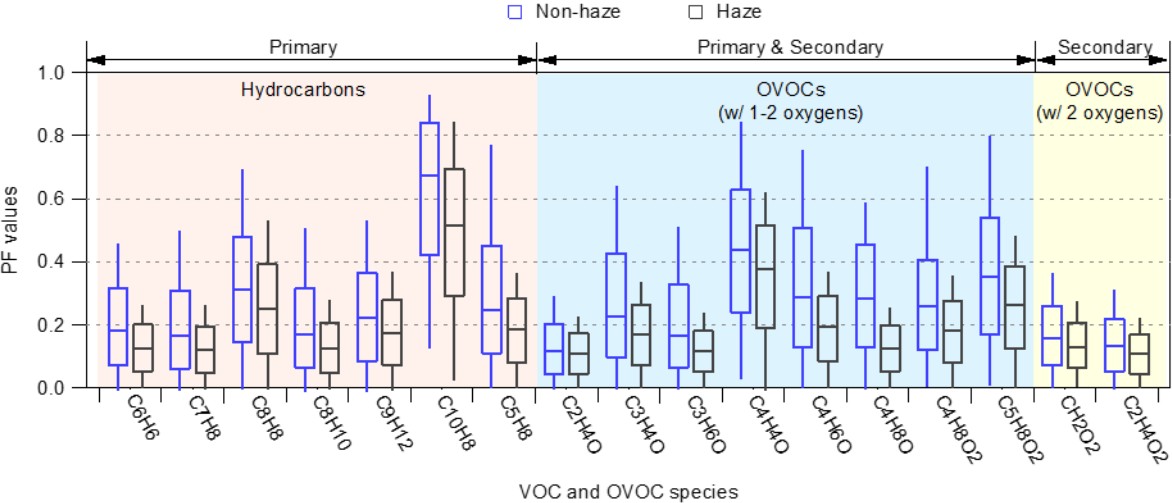

**Figure 6.** Peak fraction (PF) values for common VOC species measured on the 4th Ring Road in Beijing. Data covered from 9 a.m. to 4 p.m. during the measurement period. The box and whisker plots show the median, 75th and 25th percentiles, and 90th and 10th percentiles of PF values calculated for each data point.

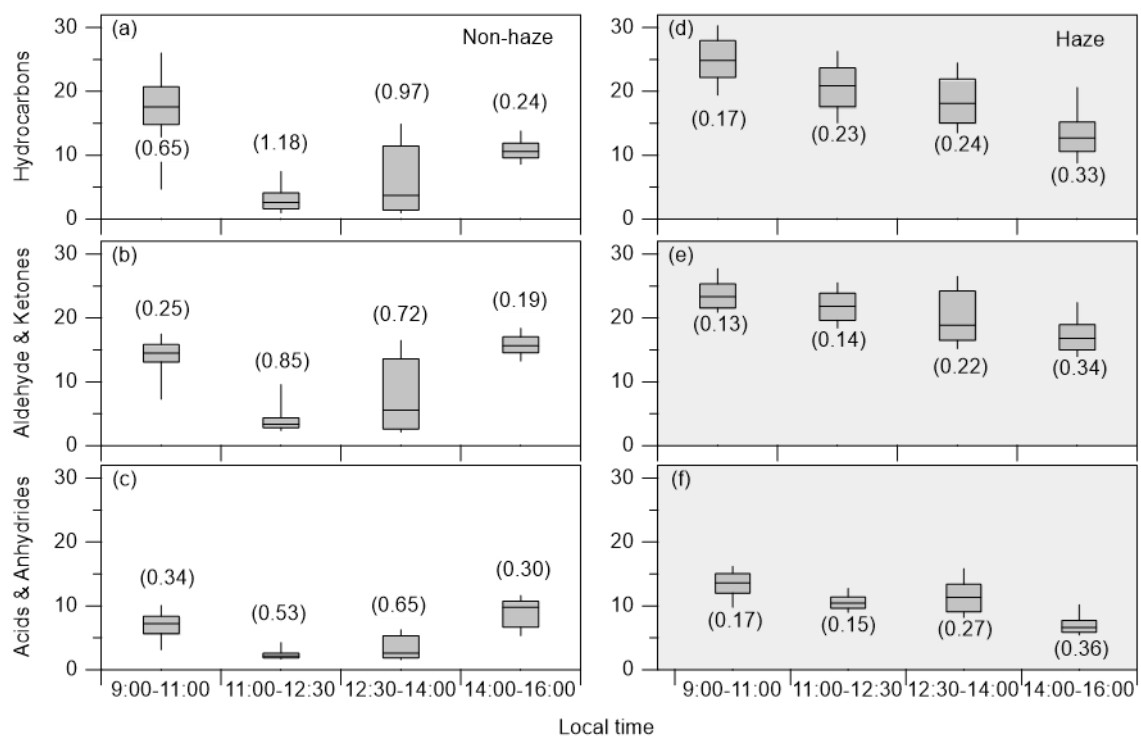

**Figure 7.** Spatial-temporal distributions of the mixing ratios (Unit: ppbv) of $\sum$ hydrocarbons, $\sum$ (aldehydes and ketones), and $\sum$ (acids and anhydrides) measured during the non-haze and haze days. The box and whisker plots show median, 75th and 25th percentiles, 90th and 10th percentiles of all the data points. The numbers in parentheses represent the CV values. Data covered from 9 a.m. to 4 p.m. during the measurement period.

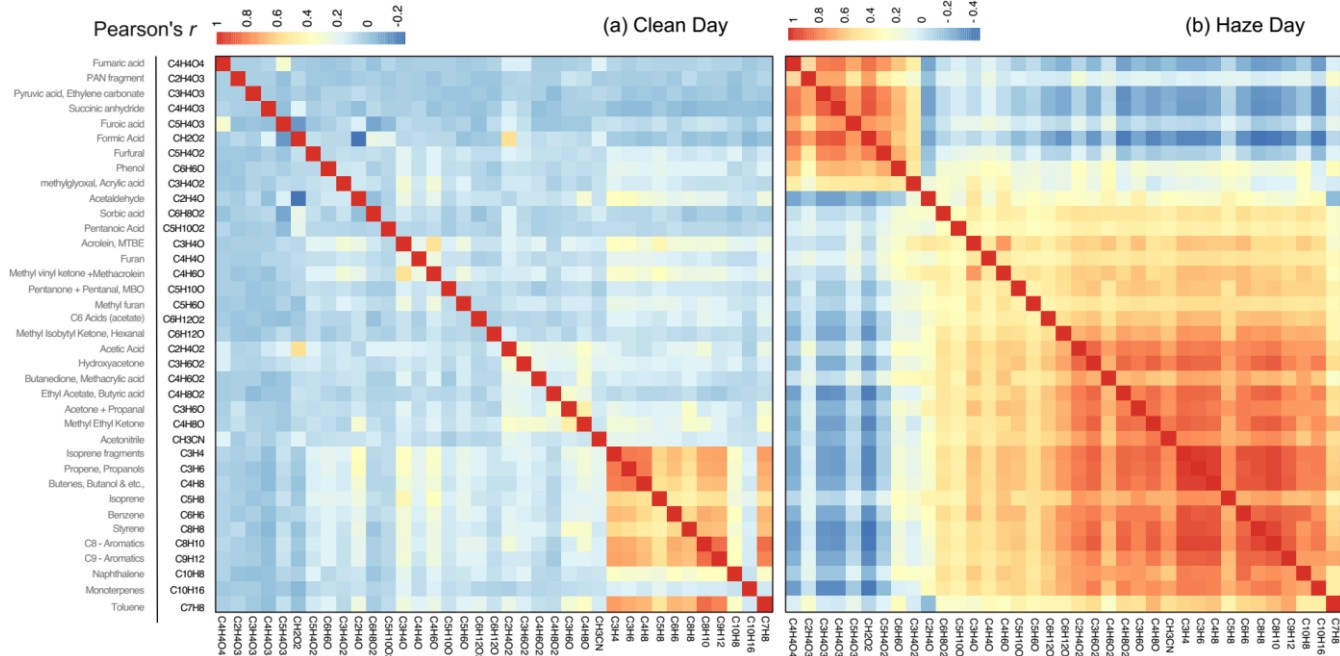

**Figure 8.** Correlation heatmaps for the concentrations of the main VOCs and OVOCs measured during (a) the clean day on 18 November 2018 and (b) the haze day on 14 November 2018.