# Peer review of "Spatial variability of air pollutants in a megacity characterized by mobile measurements"

_Atmospheric Chemistry and Physics, 2021_

## Author Comment (AC1)

**Response to reviewers**

Reviewer comments are in black *italic* type. Author responses are indented and in normal font labeled with [R]. Line numbers in the responses correspond to the revised manuscript with track-changes. Modifications to the manuscript are in *italics*.

*Reviewer #1*

*Comments:*

*This manuscript describes mobile measurements of PM mass and composition, inorganic gases, and organic vapors on haze and non-haze days in Beijing. I like the study design, which focuses on quantifying the broad spatial patterns by repeatedly driving a ring road. This is in contrast to many previous mobile sampling studies that focused on obtaining neighborhood-level details at high spatial resolution.*

*However, I have some major criticisms that need to be addressed.*

[R0] We thank the reviewer for the valuable feedback and constructive suggestions. Detailed responses are given below.

*Specific comments:*

*(1) Amount and representativeness of data: The analyses (all figures except Fig 4) rely on only two days of data (November 14 and 18, 2018). Additionally, the authors primarily discuss midday concentrations on those days. For example, Fig 1 show data from the midday drives between 11:00 am and 12:30 pm local time. Since each drive takes around 70 minutes, this means that the majority of the analysis focuses on one or two drives on each of two days. The authors claim some large conclusions (they imply that their results are representative of all haze days and all clean days). They therefore need to show more than a small slice of data on two days. The results they present here are for two days, and therefore not necessarily representative of broader conditions in Beijing. A revised version of the manuscript should include analysis from multiple clean and haze days to get a better sense of how robust the results are.*

[R1] As replied in [R4] of Reviewer #3, the day-to-day or diurnal variations of particle composition clearly present during the mobile campaign (Figure S3). Averaging the data for the whole measurement period or all clean days would smooth out the spatial variability. We therefore only presented the noon cycles to visualize the spatial variabilities of pollutants as examples. The spatial-distribution graphs are only for two days but the conclusions are made based on all the data measured during the campaign (and even data from another study in 2021).

To enrich the discussion, we have added a new graph as Figure 2. This graph shows the CV (i.e., spatial variability) distributions of all clean-day cycles vs. the haze-day cycles for the mass fractions of major particle components. Despite of the day-to-day variations, the clean-day CV values are significantly greater than the haze-day for all time periods, supporting greater spatial variabilities of aerosol composition during the clean days. We only had one haze-day data during the 2018 campaign. To support the haze-day results, we have added the haze-day results from another campaign in 2021 in Beijing in Figure 2 and Figure S10. The data also show homogeneous distributions of particle composition and

featured correlation heatmaps of VOCs and OVOCs during the haze event that are similar to the results herein. Related discussion is added in Line 188-195 and Line 307-309. The spatial-temporal distributions of VOCs are also presented in Figure 4 (now Figure 5), and we have already discussed it in the main text.

*While writing this review I looked up a 2018 calendar. November 14 was a Wednesday, and November 18 was a Sunday. I am unfamiliar with the typical Chinese workweek or people's activity patterns in Beijing, but it seems like there is a good chance that most of this paper's analysis compares a single working day to a single non-working day.*

[R2] We thank the reviewer for the good suggestion. But industrial or work activities do not vary much in China over the week compared with those in western countries. As indicated by the satellite observations of the tropospheric vertical column density of $NO_2$ and from the near-surface observations of $NO_x$, the weekend effect is insignificant in China (Hayn et al., 2009; Wang et al., 2014). Previous measurements for air pollutants in Beijing show some weekend-weekday differences but within the measurement uncertainties (Sun et al., 2015; Liu et al., 2020). Haze conditions are often associated with meteorological conditions that favor the accumulation of pollution (e.g., stagnant and humid conditions).

*(2) Interpretation of spatial homogeneity on the haze day: The authors need to provide readers with a better sense of meteorological conditions on the clean versus haze days, and how those conditions relate to their interpretation of the mobile measurements. My assumption is that the haze days have low wind speed and perhaps a low mixing height, whereas the non-haze days are windier and better mixed. That seems to be the case from the data shown in Figure S3, but the authors need to include some of that context in the manuscript. Since the haze day has lower wind speed and presumably poorer mixing, I would expect significant spatial variability, especially for primary emissions. I might even expect larger spatial gradients on haze than non-haze days because of poor dispersion. Instead, the authors explain the more homogeneous conditions on the haze day as a result of "regional transport." That doesn't make sense to me as an explanation, since the haze day seems to be a case of stagnant air where local emissions are trapped.*

[R3] We agree with the reviewer that the haze days usually have low wind speed and perhaps a low mixing height and thus the local emissions are likely accumulated more locally. However, this does not mean greater spatial heterogeneity for the mobile measurements because the on-road measurements sample air from both urban background and instantaneous plumes. The haze pollution in NCP usually develops regionally, transports to Beijing from the south, and linger in urban Beijing for days before the northwesterly/northeasterly wind with high speed blows away the pollution (An et al., 2019). Studies show that regional transport could contribute 60-70% of $PM_{2.5}$ during severe haze events in Beijing. When background air makes a major contribution to the on-road concentration of the pollutants, the impacts of accumulated local emissions on spatial distributions are perhaps reduced and spatial homogeneity presents for those pollutants. Similar to our study in Beijing, a study in Zurich shows that more than half of $PM_1$ measured in Zurich during winter are not from local emissions due to thermal inversions, resulting in a lower local/measured ratio and a rather uniform distribution of pollutant concentrations and particle composition throughout the whole Swiss plateau region. To

clarify, we have revised the discussion in Line 178-187 as follows: "*Although stagnant conditions facilitate the accumulation of local emissions (e.g., vehicle emissions on the road), over 60% of the PM$_{2.5}$ mass in Beijing can be contributed by regional transport during the winter haze episodes (Sun et al., 2014; Wu et al., 2021). The predominant contribution of regional transport suggests similar sources of PM$_{2.5}$ in Beijing. Similar particle composition suggests a spatial chemical homogeneity at least on the megacity scale in terms of gas-to-particle equilibrium or partitioning as well as the heterogeneous or particle-phase production. The north-south difference in mass concentration is perhaps driven by the differences in atmospheric dilution on the intracity scale (Sun et al., 2016; Chen et al., 2020). The uniform spatial distributions of PM composition under haze conditions are similar to the observations in the metropolitan area of Zurich when thermal inversions occur over the Swiss plateau and secondary pollution is built up regionally (Mohr et al., 2011), highlighting stagnant metrological conditions as one of the key drivers of the city-scale chemical homogeneity*.*"*

***The local emissions seem to be significant. Figure 4 shows that there are strong enough local emissions on the clean day to replenish pollutant concentrations after the boundary layer rises in the morning (e.g., hydrocarbon concentrations are higher from 12-14 and 14-16 than from 10-12). Thus, if emissions were similar on the two days, one would expect a larger daytime increase in concentrations, not a flat profile. If the haze day was a non-working day (Sunday, see comment above), emissions would be very different, and would have a major impact on the temporal patterns.***

[R4] As replied in [R6] for Reviewer #3's comments, the on-road measurements of hydrocarbons are largely affected by instantaneous vehicle plumes. Therefore, the hydrocarbon measurements herein do not represent urban background conditions. The greater concentrations of hydrocarbon in the afternoon than from 10-12 suggest less vehicle plumes that the mobile measurements captured form 10-12. This may be explained by the less traffic volume on the road. During the haze day, the stagnant condition may favor the mobile measurements to capture the high emitting plumes from 10-12 and therefore shows a rather flat profile. By contrast, for VOC and OVOCs that vehicles are not a significant source, their concentrations are affected by urban background concentrations. To clarify, we have revised the text in Line 279-294 as follows: "*The on-road measurements of hydrocarbons are largely affected by instantaneous vehicle plumes. The greater concentrations of hydrocarbon in the afternoon (2:00 p.m. to 4:00 p.m.) than in the earlier period (11:00 a.m. to 2:00 p.m.) suggest that the mobile measurements captured less vehicle plumes, which is consistent with the less traffic volume on the road. Their concentrations decrease first as the boundary layer develops, and then increase in the afternoon as the pollution accumulates in the boundary layer under non-haze conditions. Under non-haze conditions, the spatial variabilities of hydrocarbons vary significantly during the day. Their CV values are high in the morning and low in the afternoon. It is likely that the photochemistry and the better mixing conditions in the afternoon smooth out some of the spatial variabilities caused by on-road vehicle emissions (Mellouki et al., 2015; Karl et al., 2018). By contrast, their concentrations keep decreasing during the day under haze conditions, and the greater day-time concentrations of $\sum$ hydrocarbons than during the clean days are plausibly driven by the greater contribution of regional transport to on-road air and stagnant meteorological conditions that favour the accumulation of on-road*

*vehicle plumes. Under non-haze conditions, the spatial variabilities of hydrocarbons vary significantly during the day. Their CV values are high in the morning and low in the afternoon. It is likely that the photochemistry and the better mixing conditions in the afternoon smooth out some of the spatial variabilities caused by on-road vehicle emissions. Under haze conditions, the spatial variability of hydrocarbons is slightly greater in the afternoon, which is probably because of the change of regional transport direction in the afternoon.*"

***(3) With the exception of Figure 4, the authors do not show any temporal variations. I would expect that there is a lot to learn from comparing spatial patterns at different times of day (e.g., morning rush versus midday). Not showing this data in more detail seems like a major missed opportunity.***

[R5] The temporal variations of the particle composition and the concentrations of gas pollutants and VOCs in Beijing have been investigated in tremendous studies and their sources have been extensively studied. We focus here only the spatial variabilities of these pollutants and their broad implications. With the addition of the new Figure 2 and its related discussions, the spatial patterns at different times are discussed in more detail in the revised version.

*Additional comments:*

*(1) Figure 1a and 1d show the spatial variation of PM1 concentrations on two days. This figure is supposed to show that there is more variability on the clean day, however that is not obvious given the scaling of the symbols. The two days both look homogenous to me.*

[R6] We have adjusted the lay out of the composition pies in the revised Figure 1 (below) to visualize the heterogeneity better. The pies are different along the 4th Ring Road in Figure 1a,b,c but rather uniform in Figure 1 d,e,f. Table 1 lists the CV values of the mass concentrations and the mass fractions of NR-PM$_{2.5}$ components, providing quantitative information for their spatial variabilities.

[Figure]

*(2) Lines 129-130 note that most of the OA spatial variability on the clean day is due to variations in POA. However, the CV for OOA mass concentration (0.76) is similar to the CV for HOA (.79). This suggests that OOA is also variable. Though, as the authors note,*

***I would expect OOA to be more spatially homogeneous. Perhaps this high CV for OOA points to some misapportionment of other OA types as OOA.***

[R7] We agree with the reviewer that OOA also shows a great spatial variability. But we don't think this is because of the misapportionment of POA as OOA. As described in Line 106-108, BBOA or CCOA were not resolved in this data set and were perhaps mixed with OOAs. Their contributions to OA are however expected to be small because of the emission control actions according to the previous results (Zheng et al., 2020; Duan et al., 2020). OOA can be contributed by many precursors and processes. It is not surprised to see a great spatial variability. We have clarified this part in Line 143-150 as follows: "*The spatial variations of the OA mass are attributed to* both of POA and OOA. As shown in Fig. *1c, the mass fractions of POA factors such as HOA and COA show a large spatial heterogeneity with hot spots (mass fraction > 60%) in various segments of the 4th Ring Road. These hot spots are plausibly contributed by exhaust plumes from on-road vehicles and nearby restaurants that have not yet well mixed with urban background air. Similarly, the measurements in Pittsburgh show a significant spatial heterogeneity of primary carbonaceous components such as* HOA, COA, and BC (Gu et al., 2018). The Pittsburgh study show less spatial variabilities of OOAs, whereas the CV value for the OOA concentration are high in Beijing during the clean day. This is perhaps because the precursors and formation pathways of OOAs are more complicated in Beijing than in Pittsburgh (Li et al., 2021; Yang et al., 2019)*".

***(3) Fig 4 - Make it clean which panels are haze versus clear days. I assume that grey shading is for the haze days.***

[R8] We have revised Figure 4 (now Figure 5) accordingly.

***(4) Fig 4 - how many days are in each plot? Please be clear about how much data is being shown.***

[R9] There are 7 non-haze days and 1 haze day of mobile measurements in winter in 2018. We have added Section A in the supplement and revised Figure S3 to show the mobile measurement periods.

**Reference**

An, Z. S., Huang, R. J., Zhang, R. Y., Tie, X. X., Li, G. H., Cao, J. J., Zhou, W. J., Shi, Z. G., Han, Y. M., Gu, Z. L., and Ji, Y. M.: Severe haze in northern China: A synergy of anthropogenic emissions and atmospheric processes, Proc. Natl. Acad. Sci. U. S. A., 116, 8657-8666, https://doi.org/10.1073/pnas.1900125116, 2019.

Duan, J., Huang, R. J., Li, Y. J., Chen, Q., Zheng, Y., Chen, Y., Lin, C. S., Ni, H. Y., Wang, M., Ovadnevaite, J., Ceburnis, D., Chen, C. Y., Worsnop, D. R., Hoffmann, T., O'Dowd, C., and Cao, J. J.: Summertime and wintertime atmospheric processes of secondary aerosol in Beijing, Atmos. Chem. Phys., 20, 3793-3807, https://doi.org/10.5194/acp-20-3793-2020, 2020.

Hayn, M., Beirle, S., Hamprecht, F. A., Platt, U., Menze, B. H., and Wagner, T.: Analysing spatio-temporal patterns of the global NO2-distribution retrieved from GOME satellite observations using a generalized additive model, Atmos. Chem. Phys., 9, 6459-6477, https://doi.org/10.5194/acp-9-6459-2009, 2009.

Liu, Y. F., Song, M. D., Liu, X. G., Zhang, Y. P., Hui, L. R., Kong, L. W., Zhang, Y. Y., Zhang, C., Qu, Y., An, J. L., Ma, D. P., Tan, Q. W., and Feng, M.: Characterization and sources of volatile organic compounds (VOCs) and their related changes during ozone pollution days in 2016 in Beijing, China, Environ. Pollut., 257, 12, https://doi.org/10.1016/j.envpol.2019.113599, 2020.

Sun, Y. L., Wang, Z. F., Du, W., Zhang, Q., Wang, Q. Q., Fu, P. Q., Pan, X. L., Li, J., Jayne, J., and Worsnop, D. R.: Long-term real-time measurements of aerosol particle composition in Beijing, China: seasonal variations, meteorological effects, and source analysis, Atmos. Chem. Phys., 15, 10149-10165, https://doi.org/10.5194/acp-15-10149-2015, 2015.

Wang, Y. H., Hu, B., Ji, D. S., Liu, Z. R., Tang, G. Q., Xin, J. Y., Zhang, H. X., Song, T., Wang, L. L., Gao, W. K., Wang, X. K., and Wang, Y. S.: Ozone weekend effects in the Beijing-Tianjin-Hebei metropolitan area, China, Atmos. Chem. Phys., 14, 2419-2429, https://doi.org/10.5194/acp-14-2419-2014, 2014.

Zheng, Y., Cheng, X., Liao, K. R., Li, Y. W., Li, Y. J., Hu, W. W., Liu, Y., Zhu, T., Chen, S. Y., Zeng, L. M., Worsnop, D., Chen, Q., and Huang, R. J.: Characterization of anthropogenic organic aerosols by TOF-ACSM with the new capture vaporizer, Atmos. Meas. Tech., 13, 2457-2472, https://doi.org/10.5194/amt-13-2457-2020, 2020.

---

## Author Comment (AC2)

**Response to reviewers**

Reviewer comments are in black ***italic*** type. Author responses are indented and in normal font labeled with [R]. Line numbers in the responses correspond to the revised manuscript with track-changes. Modifications to the manuscript are in *italics*.

*Reviewer #3*

*Comments:*

*The paper reported the on-road mobile measurement results in megacity in China. It is interesting that homogeneous and heterogeneous spatial distributions were observed respectively for haze and clean days. The fine spatial resolution measurement provided a lot of information on localized sources, which is potentially useful for the development of future pollution control strategies. Overall, the paper is well written and logically organized. High-spatial resolution measurements is important yet scarce in China. As one of the pioneering studies in China, I recommend the paper be published subject to minor revision.*

 [R0] We thank the reviewer for the valuable feedback and constructive suggestions. Detailed responses are given below.

*Specific comments:*

*1. Line 85, both mass resolution and time resolution of the ToF-ACSM sampling should be provided.*

 [R1] We have added those information in Line 93-97 as follows: *"Gas pollutants were detected by gas analyzers including $NO_2$ (Teledyne, T500U), $NO$-$NO_x$ (Ecotech, EC9841A), $SO_2$ (Ecotech, EC9850A), CO (Ecotech, EC9830A), and $O_3$ (Ecotech, EC9810A) with a time resolution of 2 s. The chemical composition of NR-$PM_{2.5}$ was measured by an Aerodyne time-of-flight aerosol chemical speciation monitor (TOF-ACSM) with $PM_{2.5}$ lens and a capture vaporizer with a time resolution of 40 s and a mass resolution of about 400 (Zheng et al., 2020)".*

*2. Line 94 and Line 100, I don't think this is a good way to describe how the PMF results were derived and how the instruments were run during the campaign. Although experimental details had been published in the papers from the same group, readers may not have read the other ones and it is not their duty to do so. As an independent submission, at least all the necessary experimental details should be provided in SI to aid understanding of the whole manuscript.*

 [R2] We have added Section A as well as Figures S11 and S12 in the Supplement for the experimental details and the PMF analysis. The text in Line 104-106 has also been revised to guide the readers for the supplementary material.

[Figure]

Figure S11. (a) Mass spectra and (b) time series of the OA factors identified by PMF (5-factor solution).

[Figure]

Figure S12. PMF diagnostics for $Q/Q_{exp}$, variance, and residuals. Residuals are shown for the example noon cycle during the haze day.

***3. Line 125: The authors run the mobile lab on the highway, which is largely affected by the on-road vehicle emissions. Although self-contamination from the exhaust of the mobile lab could be eliminated, I'm not sure whether the data could represent the characteristic the specific area as shown on each pie in Figure 1. In another word, if the mobile lab was run on the road several meters away from the highway, would similar composition distributions be derived?***

[R3] The sampling inlets were installed at the top front of the vehicle, 3.4 m above the ground (Figure X1). The wind speed was 0.5-2 m s$^{-1}$ and sometimes 4-6 m s$^{-1}$ during the

measurement period (Fig. S3). When the mobile lab ran for cycles on the 4th Ring Road, the $PM_{2.5}$ measurements by TOF-ACSM (40 s) may roughly represent a maximum area of 0.16 km$^2$ upwind (e.g., the wind is persistently perpendicular to the mobile path at a speed of 6 m s$^{-1}$). This means a bigger footprint of our measurements than the stationary measurements on the roadside (e.g., several meters away from the highway). By contrast, the measurements of gas pollutants (2 s) represent a rather small area.

Similar composition distribution between road-side and on-road measurements may be derived for pollutants that are well mixed in the urban background and not affected by vehicle emissions (Gentner et al., 2017). For example, the particle compositions measured on the 4th Ring Road were similar to those measured by a long time-of-flight aerosol mass spectrometer (LTOF-AMS) at the PKU roof station (Figure S3). To clarify, we have added some discussion in Line 171-177 as follows: "*The sampling inlets of the PKU mobile lab are located at 3.4 m above the ground, which may sample air from both of urban background and instantaneous plumes. The 40-s $PM_{2.5}$ measurements by TOF-ACSM may roughly represent a maximum area of 0.16 km$^2$ upwind when the mobile laboratory was run on the 4th Ring Road by cycles. The similar chemical composition along the road suggests relatively homogeneous spatial distributions of the mass concentration and composition of NR-$PM_{2.5}$ across the city under haze conditions. This is supported by the fact that the particle composition observed at the PKU roof site was similar to our mobile measurements (Figure S3)*". Figure S3 is also revised with the LTOF-AMS results.

[Figure]

| Z (m) | 0.16 | 0.27 | 0.39 | 0.47 | 0.51 | 0.89 | 1.09 |
|---|---|---|---|---|---|---|---|
| V (m/s) | 6.71 | 10.71 | 15.59 | 15.57 | 15.55 | 15.48 | 15.50 |

**Figure X1.** The wind field in front of the PKU mobile laboratory at a speed of 50-60 km h$^{-1}$ modeled by FLUENT. The sampling height of Z refers to the height above the vehicle. The sampling inlet of PKU mobile laboratory was located at Z=0.4 m.

[Figure]

**Figure S3.** Time series of (a) temperature and relative humidity (RH), (b) wind speed (WS) and wind direction (WD), (c) NO, $NO_2$, and $O_3$, (d) CO and $SO_2$, (e) $PM_{2.5}$ mass concentration and chemical composition of NR-$PM_1$ measured by a long time-of-flight aerosol mass spectrometer (LTOF-AMS) at the PKU campus roof site during the entire mobile campaign in the winter of 2018. (f) and (g) particle composition of NR-$PM_{2.5}$ measured by TOF-ACSM in the mobile laboratory and particle composition of NR-$PM_1$ measured by LTOF-AMS at the PKU campus roof station during the time period of a 4th Ring Road cycle on November 14 (marked in grey in (e)), respectively. The yellow-shaded periods represent the periods having the mobile measurements.

*4. Lines 125-145, it is interesting that on clean days great spatial variability of aerosol components was observed. What about the daily variation? I'm curious whether the observed spatial variation can well represent the local emission. Also, why the authors specifically present the results of the noon cycles instead of the average of the whole cycles for one day or during all clean days' sampling since the campaign lasted for around 2 weeks.*

[R4] We have added the results of the chemical composition of non-refractory submicron particles measured by the LTOF-AMS at the PKU roof station to Figure S3. As shown in Figure S3, the day-to-day or diurnal variations of particle composition clearly present. Averaging the data for the whole measurement period or all clean days would smooth out the spatial variability. We therefore only presented the example noon cycles in Figure 1. To support our conclusion that the spatial variability of aerosol composition is greater during the clean days than during the haze day, we have added a new graph as Figure 2. This graph shows the CV (i.e., spatial variability) distributions of all clean-day cycles vs. the haze-day cycles for the mass fractions of major particle components. During the 2018 mobile campaign, we only had one haze-day data. In the revised manuscript, we have added another haze-day data that were collected on 21 January 2021 in Beijing. Despite of the day-to-day variations, the clean-day CV values are significantly greater than the haze-day values for all time periods.

[Figure]

**Figure 2.** The CV values for the inorganic and organic mass fraction in NR-PM$_{2.5}$ for all cycles during the mobile campaign. The box plots show the 75th, median, and 25th percentiles.

**5. Line 164: megacity scale? Or the authors meant the regional scale?**

[R5] Yes, we meant megacity scale. The severe winter haze is typically a regional event. But we have only measured in Beijing and have no data to tell whether the particle composition were similar outside urban Beijing. To be clear, we have revised the text as follows: "*The similar particle composition may suggest a chemical homogeneity at least on the megacity scale*" in Line 171.

**6. Line 248: Why hydrocarbons accumulated in the afternoon (12:00pm-14:00pm)? Hydrocarbons should decrease during the noon time because of photochemical consumption as observed from on-site measurements in literature.**

[R6] We agree with the reviewer that photochemical consumption may lead to a noontime valley of hydrocarbon concentrations as observed in urban background site. The on-road measurements of hydrocarbons are however largely affected by instantaneous vehicle plumes. Therefore, the measurements herein do not represent urban background conditions for hydrocarbons. As shown in Figure 4, the median concentrations during 12:00-14:00 are lower than the morning concentrations but the data span in a wide range.

To clarify, we have revised the text in Line 279-290 as follows: "*The on-road measurements of hydrocarbons are largely affected by instantaneous vehicle plumes. The greater concentrations of hydrocarbon in the afternoon (2:00 p.m. to 4:00 p.m.) than in the earlier period (11:00 a.m. to 2:00 p.m.) suggest that the mobile measurements captured less vehicle plumes, which is consistent with the less traffic volume on the road. Under non-haze conditions, the spatial variabilities of hydrocarbons vary significantly during the day. Their CV values are high in the morning and low in the afternoon. It is likely that photochemistry and better mixing conditions in the afternoon smooth out some of the spatial variabilities caused by on-road vehicle emissions (Mellouki et al., 2015; Karl et al., 2018). By contrast, their concentrations keep decreasing during the day under haze conditions, and the greater concentrations of $\sum$ hydrocarbons than during the clean days are plausibly driven by the greater contribution of regional transport to on-road air and stagnant meteorological conditions that favour the accumulation of on-road vehicle plumes*"

**7. From the discussion in Section 3.3, it seems variations of VOCs and OVOCs species are predominantly driven by on-road vehicles or high-emitting plumes. The running cycles on the 4th Ring Road cover different regions characterized by different functions, such as industrial area, residential area, etc., yet the VOC characteristics in different regions were not discussed in detail except vehicle emission. Could more information on local sources for different regions be derived from the measurements? After all, mobile emission is not the only emission source.**

[R7] Yes, we agree with the reviewer that local sources can affect the on-road mobile measurements. For example, as we mentioned in Line 246 that the high T/B ratios in the south region of the 4th ring road may be explained by industrial plumes (e.g., from chemical plants, painting processes, or constructions involving evaporation emissions). Cooking exhaust plumes present as well as indicated by the COA hotspot in Figure 1c. We have clarified in Line 250-251 that mobile emissions are not the only source that influence the on-road air.

**8. Line 540: Legend, non-haze and haze days should be denoted in Figure 4.**

[R8] We have revised this figure (now Figure 5) to clarify the non-haze and haze-day results.

**Reference**

Gentner, D. R., Jathar, S. H., Gordon, T. D., Bahreini, R., Day, D. A., El Haddad, I., Hayes, P. L., Pieber, S. M., Platt, S. M., de Gouw, J., Goldstein, A. H., Harley, R. A., Jimenez, J. L., Prevot, A. S. H., and Robinson, A. L.: Review of urban secondary organic aerosol formation from gasoline and diesel motor vehicle emissions, Environ. Sci. Technol., 51, 1074-1093, https://doi.org/10.1021/acs.est.6b04509, 2017.

---

## Author Response (AR2)

**Response to reviewers**

Reviewer comments are in black *italic* type. Author responses are indented and in normal font labeled with [R]. Line numbers in the responses correspond to the revised manuscript without track-changes. Modifications to the manuscript are in *italics*.

*Reviewer #1*

*I thank the authors for the responses to my first set of comments and their edits to the manuscript. However, I do not feel like they have done enough to warrant publication, and I continue to have major issues with this manuscript. Overall comment: I think that the authors have collected an interesting dataset, and the manuscript gives some glimpses into interesting conclusions that might be reached from the data (e.g., I am particularly interested in the spatial variation in inorganic PM2.5). However, there does not seem to be a coherent story. I instead got the impression that there were a set of discrete explanations for each chunk of the data (e.g., high emitting vehicles or regional transport), even if that explanation did not hold up for another part of the dataset. I think the authors need to step back and tell a coherent story about the full dataset. If that is not possible (and it may not be!), they should be straightforward about the limitations of the dataset and the analyses presented in the paper. For example, the small number of haze days (there seem to be two, but the authors are not forthcoming about this) is a limitation of the dataset. That is fine if there are only two haze days, but right now I feel like some of the details are being downplayed, and that makes me wonder if other aspects of the data collection and analysis are not being shown. Hopefully the comments below help to flesh out the concerns listed in this overall comment.*

[R0] We thank the reviewer for the valuable feedback and constructive suggestions. We have major changes to the manuscript to address the reviewer's comments. Detailed information about the data collection and analysis have also been added to the main text and the supplementary. We think the results are presented in a much better way in the revised manuscript and the story behind has been clarified. Detailed responses are given below.

*Major comment 1: My first set of comments criticized the manuscript for relying on what seemed to be a single transit of the 4th-ring road on two separate days (one haze and one non-haze). I don't feel like that comment was adequately addressed. In their response, the authors state "Averaging the data for the whole measurement period or all clean days would smooth out the spatial variability." However, by relying on only one or a few sampling passes to make their point, the authors risk overdue influence by quasi-random events such as driving near high emitting vehicles. Spatial aggregation over multiple sampling drives is needed to remove the influence of these events and to reveal the longer-term spatial patterns. Averaging over multiple drives is critical if the authors want to draw general conclusions from the mobile sampling. The influence of quasi-random high emission events is shown graphically by Apte et al (2017). Other papers, including Gu et al (which the authors cite) address the issue of "how many" mobile sampling passes are needed to build robust spatial patterns with mobile sampling.*

[R1] We agree with the reviewer that averaging is necessary to derive a longer-term

spatial pattern for general conclusions from the mobile sampling. Our original focus was mainly on the spatial variations, and therefore only one driving cycle of pollutant distributions is shown for non-haze vs haze conditions. The presented spatial variations might be biased by quasi-random emission events. In the revised manuscript, we have averaged all drives from 9 AM to 4 PM over 8 non-haze days to derive the non-haze spatial patterns. The haze-day case is limited to 1-day average of the data, which has been clarified in the main text about the data limitation. But we have discussed some of the key features of the data in Line 117-138. The haze day herein represents a typical winter-haze event in the later high relative humidity stage. The findings from the haze-day spatial patterns are confirmed by the analysis of another haze-day in the 2021 mobile campaign. The revised manuscript now focused on discussing the general spatial patterns. Sections are re-organized and figures are replaced.

***Major comment 2: The authors added Figure 2 to try and address my comment about temporal or drive-to-drive variation. However, this figure generates more questions than answers for me. I don't understand what Figure 2 shows. There are box plots, but how are they constructed? Is there one CV calculated for each time around the ring? One for each sampling day? Additionally, why is the organic PM normalized to PM$_{2.5}$ mass, but the inorganic components are normalized to the sum of inorganics? Why not normalize everything to PM$_{2.5}$ mass?***

[R2] The original Figure 2 shows the box plots of CV for each drive cycle on the 4[th] Ring Road for all the cycles in the 8 non-haze days and 2 haze days. Because the revised manuscript now focuses on the spatial pattern of mean concentrations, this figure is no longer necessary and has been deleted from the main text. We have now used the magnitude of concentration variation and the CV values of the spatial patterns of mean concentrations to discuss about the spatial variability. Tremendous work has been done in previous studies to investigate the temporal variations of common gas pollutants and aerosol species. Temporal variations of these pollutants are not our focus herein. We therefore only discussed about the temporal variations for VOC and OVOCs in Sect. 3.3 of the revised manuscript.

***Major comment 3: I am still not convinced by the author's reasoning for a lack of spatial heterogeneity on the haze days. They try to explain this away with a hand-waving nod to "regional transport." However, on stagnant haze days, local plumes do not disperse, and their impacts should be larger. For example, Lines 136-138 attribute HOA hotspots on the clean day to high emitting vehicles. If occasional high emitting vehicles are truly the source of the hotspots, the authors should detect these (or similar) hotspots on the haze days. A lack of these hotspots would seem to undermine the conclusion that pollutants are more homogeneous on the haze day. Rather, it would mean that the mobile lab simply passed fewer high emitting vehicles on the haze day. Another example of stagnant plumes on haze days: Lines 217-233 discuss high on-road emissions and titration of O3 on the highway. This is evidence of a strong emission source and spatial gradients associated with that source. And on haze days the data should see vehicle plumes, unless traffic volumes are vastly different on haze and non-haze days (or perhaps bad luck passing high emitting vehicles on the haze days).***

[R3] We agree with the reviewer that local plumes are less dispersed much under stagnant conditions. The revised manuscript presents the averaged spatial patterns, which are much clearer about the enhanced impacts of local sources. For example, in Line 231, we state that "*The mean mixing ratios of CO were however greater than the non-haze day ratios, indicating accumulated pollution*". In Line 270-271, we state that "*Hot spots of HOA and COA became more evident, which is consistent with the less-dispersed primary emissions under stagnant conditions (Figure 4)*". While the spatial variabilities for all pollutants were significant for all pollutants during the non-haze days, the spatial variability for secondary aerosol species (e.g., OOA, sulfate, nitrate, and ammonium) and OVOCs have been largely reduced. The haze in NCP is usually developed regionally, meaning that the polluted air mass travels and would become more polluted when it suspends in urban areas to accumulate local emissions and secondary production under stagnant conditions. During the haze event, polluted air mass arrives and leads to significantly greater urban background concentrations for both primary and secondary pollutants. Meanwhile, secondary formation can be enhanced because of the elevated precursor concentrations during the haze day and heterogeneous and aqueous pathways for aerosol species that occur during the high-RH haze stage. The two facts drives a rather homogeneous distribution of aerosol composition because secondary species dominate the mass during the haze day. This has been clarified in the new Sect. 3.2 - "Spatial distribution and variability during the haze day".

***Major comment 4: Some information about sources or land use would be helpful. There are some general descriptions in the text, but a graphical representation would be better. Most readers are not familiar with the land use in Beijing. Linking the land uses to the observed spatial variations in a more concrete way would help drive home the conclusions of this manuscript.***

[R4] We have added Figure 1 for information about land use and vehicle emissions in the revised manuscript. Indeed, the land use information is helpful. For example, in Line 79-80, we added the following "*The 4th Ring Road is a 65-km-long urban highway that passes through residential, commercial and services, park, and transportation areas in the megacity (Figure 1a)*". In Line 163-165, We discussed as follows: "*Overall, the spatial pattern of NOx was consistent with the bottom-up emission inventory for (1) the nonuniform vehicle emissions on the 4th Ring Road and (2) high concentrations in the east segment of the 4th Ring Road where the traffic volume was high (Figure 1b and Figure S10 in the Supplement)*". In Line 192-197, we discussed about local sources as follows: "*The 40-s PM$_{2.5}$ measurements by TOF-ACSM may roughly represent a maximum area of 0.16 km$^2$ (for a mean speed of 6 m s$^{-1}$ and wind direction perpendicular to the mobile path) upwind when the mobile laboratory was run on the 4th Ring Road by cycles. The HOA hot spots are generally consistent with the locations where the traffic volume was high and the driving speed was relatively low (Figure S10 of the Supplement). The COA hot spots are consistent with the places where the 4th Ring Road passes through sparsely located residential areas (Figure 1a)*".

***Major comment 5: Fig 4 is hardly discussed in the text. Some of the OVOCs have high PFs on the non-haze days. What are possible sources? The text focuses on vehicles as the main source of spatial variation - do vehicles emit things like furoic acid?***

[R5] We have revised this figure (now Figure 6) with common VOC species likely related to primary vehicle emissions and secondary production. We have added more detailed discussions for Figure 6 in Line 253-265 as follows: "*The calculated PF for VOCs ranged from 11-67% (median) (Figure 6). High PF values were found for hydrocarbons and some OVOCs (e.g., C8H8, C10H8 and C4H4O), indicating a major contribution of transient localized sources (e.g., traffic, industrial facilities) to these species. The time series of these so-called primary species showed low baselines and sharp peaks (Figure S6). By contrast, OVOCs (i.e., with 2 or more oxygen in their formulae) that were typically considered as secondary species had low PF values (median: 11-16%) and elevated baseline contribution from photochemistry. Significance tests indicate greater PF values for the primary species during the non-haze days, meaning that the localized sources contributed more to the measured concentrations during the non-haze days than during the haze day ($p < 0.001$). During the haze day, the localized emissions should be accumulated near the source under stagnant conditions. Indeed, the peak concentrations of primary VOC species were significantly greater (e.g., ~2-4× for C6H6 and C7H8) (Figure S6). The lower PF values (by 30% for C6H6 and C7H8) during the haze day were caused by much more elevated baselines (e.g., ~9× for C6H6 and C7H8) that represent urban background affected by polluted air mass from regional transport plus gradually mixed local emissions. The mean VOC concentrations at the PKU roof site increased for about 2 times during the haze day, which agrees with the elevated baselines (Table S3 and Figure S12 in the Supplement).*"

***Major comment 6: In my first round of comments I questioned whether some of the OOA spatial variation could be the result of misapportionment. The authors responded in part with "OOA can be contributed by many precursors and processes. It is not surprised to see a great spatial variability." I disagree vigorously. Primary OA will be spatially variable because it is emitted by local sources. OOA, which requires chemical processing, would be expected to be more spatially homogeneous. (At least I would consider this the null hypothesis, and the authors would need to disprove the null, which they have not done.)***

[R6] Yes, non-perfect separation of POA and OOA by the PMF analysis may lead to misplaced spatial variability in OOA (Section A3). We have added detailed descriptions about source apportionment of OA by PMF in Section A3 of the Supplement, Figures S2-S5, and Table S1. The uncertainty of the PMF analysis has been clarified. The signal of *m/z* 44 of the PMF factors is sensitive to the rotation choice, which may introduce some uncertainty of the PMF results. During the non-haze days, the mean spatial pattern of OOA still shows moderate spatial variability (now Figure 4). In Line 197-208, we explained this as follows: "*Moreover, the mass concentrations of the sum of OOAs varied from 0 to 15 µg m⁻³. Local photochemical production of SOA is a significant source of OA in Beijing in winter, although the solar radiation is reduced (Duan et al., 2020; Lu et al., 2019a). The photochemical production depends on the distributions of SOA precursors and oxidants.*"

*In the northwest corner where hydrocarbons showed high concentrations, the OOA mass loadings were indeed high. Because the majority of the SOA precursors (i.e., intermediate volatility and semivolatile organic species from anthropogenic sources) were not measured by the PTR-Qi-ToF (Liao et al., 2021; Miao et al., 2021), it is difficult to investigate more about the OOA source. The measurements in Pittsburgh also showed a significant spatial heterogeneity of primary carbonaceous components such as HOA, COA, and BC (Gu et al., 2018). Less spatial variability presented for OOAs in the Pittsburgh study. The OA mass loadings in Pittsburgh were however much less than the loadings in Beijing. The SOA formation can be significantly more efficient and complicated under conditions of high oxidative capacity and abundant precursors in Beijing than in Pittsburgh (Lu et al., 2019a; Li et al., 2021; Yang et al., 2019). Non-perfect separation of POA and SOA by the PMF analysis may also lead to misplaced spatial variability in OOA (Section A3)*".

**Minor comments**

**Line 84-87 refer to traffic volumes and composition on the 4th ring road. This needs a reference.**

[R7] We thank the reviewer for the suggestion, and have added references. The traffic volume information is updated to the mean daily volume in November 2018.

**Paragraph starting on line 88 - the vehicle speed was typically 60 km/h and AMS sampling times were 40 s. This gives a spatial resolution of about 700 m for PM mass and composition, and should be stated.**

[R8] We thank the reviewer for the suggestion and have stated the resolution in Line 93-94 as follows "*The time resolution was 40 s, corresponding to a spatial resolution of ~0.7 km for a driving speed of 60 km h-1*".

**Line 82 notes that measurements were collected on Nov 7-21 and Jan 21. Was sampling conducted on all of the November days? How many haze and non-haze days were sampled?**

[R9] Sampling was conducted almost every day excluding some days for instrument and OFR maintenance. The effective sampling covered only 8 non-haze days and 1 haze day in 2018, which has been clarified in Line 122.

**Figure 3 would benefit from the Clean and Haze days using the same color scale for each pollutant.**

[R10] We have revised the color scales (now Figure 2).

**Line 164 - where does the value of 0.16 km^2 come from?**

[R11] Assuming the wind is persistently perpendicular to the mobile path at a mean speed of 6 m s$^{-1}$, the maximum area that the mobile measurement could represent is 60 km h$^{-1}$ × 6 m s$^{-1}$ × 40 s × 40 s = 0.16 km$^2$. We have clarified this in Line 193.

***Line 202 and 203 - concentrations for "non-haze" days are listed twice; clearly one should be haze days.***

[R12] Yes. We have revised it as follows "*For CO, the mean mixing ratio of 1.5 ± 0.7 ppmv was about three times greater than the urban background level (0.5 ± 0.3 ppm at the PKU roof site), indicating significant contributions of localized sources during the non-haze days.*" The haze-day discussion has been moved to Sect. 3.2.

***Line 287-288, when describing Fig 6, the authors state "Day-to-day variations are not included because of the possible change of sources." I do not understand what this means. Does it mean that Fig 6 is presented for a single haze and non-haze day?***

[R13] Yes, Figure 6 (now Figure 8) is presented for the haze day and a single non-haze day. The averaged spatial patterns have been discussed in the revised Section 3.1 and 3.2. Herein, we selected one clean day to compare with the haze day for correlations of VOCs.

***Figue S5 shows VOC baselines. How were these determined and were they used?***

[R14] We have described how these baselines were determined in Line 106-108 as follows: "*Baseline concentrations for each 2-s point in the 20-s smoothed data that represents were calculated as the 5th percentile concentration within a rolling window of 60 (i.e., 120 s) to represent the well-mixed urban background conditions.*"

***Not much discussion of Fig 4.***

[R15] As replied in [R5], we have revised this figure (now Figure 6) and added more discussion about this figure in Line 255-267.

***I still don't understand how stagnant conditions in the haze day lead to things being more regional. More photochemically active? But if weather is stagnant, you should see stronger plumes near sources.***

[R16] As replied in [R3], we agree with the reviewer that the stagnant conditions may enhance the impact of local plumes. Indeed, we have seen it in the data. But the particle composition was dominated by OOA, sulfate, nitrate, and ammonium during the haze day. Those secondary species had reduced spatial variability compared to the non-haze case.

---

## Author Response (AR3)

**Response to reviewers**

Reviewer comments are in black *italic* type. Author responses are indented and in normal font labeled with [R]. Line numbers in the responses correspond to the revised manuscript without track-changes. Modifications to the manuscript are in *italics*.

*Reviewer #1*

*This paper presents mobile field measurements of fine particle compositions, VOCs, and trace gases on an urban highway road in Beijing. Spatial distributions of different air pollutants under haze and non-haze conditions were investigated. Reference measurement at a fixed station representing the typical urban environment was also conducted to facilitate the analysis of spatial distribution and variability of different pollutants from mobile measurement. This is a revised paper that has been reviewed by two other reviewers. The authors' responses have clearly shown that they have made major changes to the manuscript to address the reviewer's comments. The revised paper is better organized, and the results are presented more clearly. Overall, I support the acceptance of the revised paper after some minor comments are addressed.*

[R0] We thank the reviewer for the valuable feedback and constructive suggestions. Detailed responses are given below.

*1. The author used the PKU roof site as a reference station for comparing the mobile measurement; however, there is almost no information on the reference station in the methods section.*

[R1] We have added the description about the PKU roof site in section in Line 86-93 as follows: "*Additionally, online measurements of gaseous and particulate pollutants were conducted at a roof station in the PKU campus (39.99 °N, 116.32 °E) as a reference. Temperature, RH, barometric pressure, wind speed, and wind direction were acquired by a Met One weather station (083E, 092, 010C, and 020C). Gas pollutants were measured by Thermo Scientific instruments, including CO (48i-TL), NO-NO$_2$-NO$_x$ (42i-TL), SO$_2$ (43i-TL), and O$_3$ (49i-TL). PM$_{2.5}$ mass concentrations were measured by a tapered element oscillating microbalance monitor (Thermo, TEOM 1400A). Non-refractory chemical components of submicron particles (NR-PM1) were measured by an Aerodyne long time-of-flight aerosol mass spectrometer (LTOF-AMS). The roof site is located between the 4th and the 5th North Ring Roads, representing a typical urban background environment in Beijing (Zheng et al., 2021)*".

*2. Figure 1. Panel d is missing in the figure caption.*

[R2] We have added the description for panel d in the figure caption as follows: "*(d) The average NR-PM$_{2.5}$ composition measured by the TOF-ACSM on the mobile lab on the 4th Ring Road within the distance of 1.5 km from the roof site*".

*3. Line 122-123. It would be better if the corresponding periods of mobile measurement campaigns could be explicitly shown in the time series figure (e.g. Figure S8).*

[R3] We have marked the mobile measurement periods in Figure S8.

***4. Line 130-131 and Figure S8. The wind directions are not clear in the time series figure. It will be clearer if the wind speed and direction can be depicted in the angular arrows.***

[R4] We have revised panel (b) for wind speed and direction in Figure S8.

***5. Line 159 and Line 243. Do you mean the titration of $O_3$ by NO? To consider the titration effects of $O_3$ converting NO to $NO_2$, a better approach is to compare the $O_x$ ($NO_2$+$O_3$) instead of $NO_2$ itself.***

[R5] Yes, we mean the titration of $O_3$ by NO. The mean mixing ratios of on-road $O_x$ (114.1 ppbv) were much greater than the roof-site mean (48.7 ppbv) during the non-haze periods, indicating a possible contribution of direct tailpipe $NO_2$ emissions. However, the roof site was located in the upwind direction and was less affected by urban traffic emissions. The ozone concentration at the roof site might be lower than the on-road level without titration and thus quantitative conclusion is difficult to make. To clarify, we have revised Line 167-174 as follows: "*On-road $NO_2$ can be contributed by direct tailpipe $NO_2$ emissions, NO titration, and urban background. The mean mixing ratios of on-road $O_x$ (114.1 ppbv) were much greater than the roof-site mean ratio (48.7 ppbv) during the non-haze periods, indicating a possible contribution of direct tailpipe $NO_2$ emissions, although tailpipe $NO_2$ emissions for LDGVs of National Stage III to V are expected to be low (Wu et al., 2017). On-road NO titration can also be strong (Yang et al., 2018), considering that the spatial patterns for $O_3$ and NO were anti-correlated (Pearson r = -0.43) and the mean mixing ratio of on-road $O_3$ (11.2 $\pm$ 2.2 ppbv) were over 2 times lower than the roof-site observations (25.7 $\pm$ 12.8 ppbv). Quantitative analysis on their relative contributions to on-road $NO_2$ is however difficult because the site was located in the upwind direction*".

***6. Line 170-176, from Figures 2 and 7, it seems the hydrocarbon showed the greatest spatial variability compared to the acids/anhydrides. Please check and clarify the discussions in this part.***

[R6] Yes, hydrocarbons showed the greatest spatial variability among detected VOCs. We have revised Line 180-183 as follows: "*During the non-haze days, the tentatively-assigned group of hydrocarbons showed the greatest spatial variability (by 21$\times$) among VOCs, whereas the groups of aldehydes/ketones and acids/anhydrides showed less variability (by 4$\times$ and 9$\times$, respectively) (Table S2). Secondary production is expected to contribute greatly to the latter two groups (Wang et al., 2021)*".

***7. Line 194-195. Is it possible to show a correlation plot on the hot spots of different pollutants with the traffic volume? Any relationship between the HOA with other primary tracers, like CO, NO, or some hydrocarbon?***

[R7] Unfortunately we can't obtain the real-time traffic volume data. The driving speed is not a quantitative indicator and only describes the traffic jam qualitatively. The correlations of HOA with other gaseous species that are related to traffic sources such as CO, NO, and hydrocarbons are weak ($r < 0.2$), although better than the correlations between COA (or OOA) and these gaseous species. Possible reasons for the weak correlations include different dilution or mixing between gas and particulate pollutants, different contributions of urban background, and the influence of other localized sources

to some of the pollutants. To clarify, we have revised Line 202-204 as follows: "*The HOA hot spots are generally consistent with the locations where the driving speed was relatively low (i.e., perhaps high traffic volume). We use the driving speed to indicate the traffic volume because the real-time traffic volume data weren't available (Figure S10 in the Supplement)*".

**8. Line 289-292. Do the authors mean the earlier period has fewer vehicle plumes? Please clarify the information here.**

[R8] Yes, there are usually less traffic around noon in Beijing. We have clarified this information in Line 300-304 as follows: "*The traffic volume in Beijing is usually less around noon than in the afternoon (Wu et al., 2017). Consistently, the concentrations of hydrocarbons were greater with larger variations in the afternoon (2:00 p.m. to 4:00 p.m.) than in the earlier period (11:00 a.m. to 2:00 p.m.), indicating more vehicle plumes were captured by the mobile measurements in the afternoon*".

*Reviewer #2:*

**This study utilizes detailed in-situ mobile measurements of aerosol compositions and VOCs to investigate the intracity scale variability of air pollution and its sources in the megacity of Beijing. Distinct spatial variability of air pollutants, sources and chemical processes was found for haze and non-haze conditions. Overall, the manuscript is well organized and clearly written, with the methods and interpretation being solid and convincing. I would like to recommend that it can be accepted after the following minor comments being addressed.**

[R0] We thank the reviewer for the valuable feedback and constructive suggestions. Detailed responses are given below.

**Line 99-101: BBOA and CCOA were not resolved in the PMF solution. I am wondering whether those primary OA were mixed with HOA, COA, or OOAs. Would the PTR-MS data provide additional information for biomass burning or coal burning tracers (e.g., acetonitrile)?**

[R1] BBOA and CCOA were likely mixed with HOA because of the similarity of their spectra with capture vaporizer (Zheng et al., 2020). Indeed, the PTR-MS data showed low concentrations of acetonitrile ($0.15\pm0.20$ ppbv for non-haze periods and $0.60\pm0.22$ ppbv for the haze periods) during the mobile campaign. By comparison, the average concentrations of acetonitrile are usually greater than 0.7-1 ppbv in the winter of Beijing when biomass and coal burning contributes significantly (Huangfu et al., 2021; Shi et al., 2020). We have added this information in Line 108-113.

**Line 147: "in the southeast areas" instead of "in the southeast and lowest areas"?**

[R2] We have revised the text accordingly.

**Line 170-171: The concentration variations of hydrocarbons look more significant than those of acids/anhydrides.**

[R3] Yes. As replied in [R6] for Reviewer #1, we have revised Line 180-183 for clarification.

**Line 207-208: Does non-perfect separation mean inappropriate attribution of POA to SOA or the reverse?**

[R4] As explained in Section A3 in the Supplement, the signal intensity of $m/z$ 44 is sensitive to the PMF rotation. Therefore, we may overestimate or underestimate SOA because of the rotation choice. The CV-based PMF tends to overestimate the OOA mass comparing with the SV-based analysis, meaning inappropriate attribution of POA to SOA (greater spatial variability in resolved SOA). However, there is a lack of standard methods for quantifying POA and SOA. It is hard to conclude that. To clarify, we have revised this in Line 216-219 as follows: "*Non-perfect separation of POA from SOA by the PMF analysis may also lead to misplaced spatial variability in OOA. For example, the CV-based PMF analysis may overestimate the SOA mass comparing with the traditional AMS analysis (Zheng et al., 2020), which may lead to inappropriate attribution of POA to SOA and thus more spatial variability in SOA. Uncertainty remains in the mass separation of POA and SOA (Sect. A3 of the Supplement)*".

**Line 337: The formula here should be PAN's fragment not PAN itself. How good can this ion represent PAN? Please clarify.**

[R5] Peroxyacetyl nitrate (PAN) can undergo proton transfer reaction with $H_3O^+$ ions, resulting in a protonated signal at $m/z$ 122.008. Protonated PAN subsequently undergoes a secondary ion-molecule reaction with water to form a product ion $CH_3C(O)OOHH^+$ with $m/z$ 77.023 (the ion herein) as well as fragment ions at $m/z$ 43.018 and $m/z$ 45.993 (Hansel and Wisthaler, 2000; Kaser et al., 2013; Yuan et al., 2017). The signal at $m/z$ 77.023 correlated well with PAN measured by specific PAN instruments (de Gouw et al., 2003; Kaser et al., 2013). For the high-resolution PTR-QiTOF, possible interferences are protonated peracetic acid ($C_2H_4O_3H^+$) and glycolic acid ($C_2H_4O_3H^+$) while they should have minor contributions in our measurements in Beijing (Yuan et al., 2017). We have revised the text in Line 349-352 to clarify.

**References**

de Gouw, J. A., Goldan, P. D., Warneke, C., Kuster, W. C., Roberts, J. M., Marchewka, M., Bertman, S. B., Pszenny, A. A. P., and Keene, W. C.: Validation of proton transfer reaction-mass spectrometry (PTR-MS) measurements of gas-phase organic compounds in the atmosphere during the New England Air Quality Study (NEAQS) in 2002, J. Geophys. Res-Atmos., 108, 18, https://doi.org/10.1029/2003jd003863, 2003.

Hansel, A., and Wisthaler, A.: A method for real-time detection of PAN, PPN and MPAN in ambient air, Geophys. Res. Lett., 27, 895-898, https://doi.org/10.1029/1999gl010989, 2000.

Huangfu, Y., Yuan, B., Wang, S. H., Wu, C. H., He, X. J., Qi, J. P., de Gouw, J., Warneke, C., Gilman, J. B., Wisthaler, A., Karl, T., Graus, M., Jobson, B. T., and Shao, M.: Revisiting acetonitrile as tracer of biomass burning in anthropogenic-influenced environments, Geophys. Res. Lett., 48, 10, https://doi.org/10.1029/2020gl092322, 2021.

Kaser, L., Karl, T., Schnitzhofer, R., Graus, M., Herdlinger-Blatt, I. S., DiGangi, J. P., Sive, B., Turnipseed, A., Hornbrook, R. S., Zheng, W., Flocke, F. M., Guenther, A., Keutsch, F. N., Apel, E., and Hansel, A.: Comparison of different real time VOC measurement techniques in a ponderosa pine forest, Atmos. Chem. Phys., 13, 2893-2906, https://doi.org/10.5194/acp-13-2893-2013, 2013.

Shi, Y. Q., Xi, Z. Y., Simayi, M., Li, J., and Xie, S. D.: Scattered coal is the largest source of ambient volatile organic compounds during the heating season in Beijing, Atmos. Chem. Phys., 20, 9351-9369, https://doi.org/10.5194/acp-20-9351-2020, 2020.

Wu, Y., Zhang, S. J., Hao, J. M., Liu, H., Wu, X. M., Hu, J. N., Walsh, M. P., Wallington, T. J., Zhang, K. M., and Stevanovic, S.: On-road vehicle emissions and their control in China: A review and outlook, Sci. Total Environ., 574, 332-349, https://doi.org/10.1016/j.scitotenv.2016.09.040, 2017.

Yang, B., Zhang, K. M., Xu, W. D., Zhang, S. J., Batterman, S., Baldauf, R. W., Deshmukh, P., Snow, R., Wu, Y., Zhang, Q., Li, Z. H., and Wu, X.: On-road chemical transformation as an important mechanism of NO2 formation, Environ. Sci. Technol., 52, 4574-4582, https://doi.org/10.1021/acs.est.7b05648, 2018.

Yuan, B., Koss, A. R., Warneke, C., Coggon, M., Sekimoto, K., and de Gouw, J. A.: Proton-transfer-reaction mass spectrometry: Applications in atmospheric sciences, Chem. Rev., 117, 13187-13229, https://doi.org/10.1021/acs.chemrev.7b00325, 2017.